# When LLMs Develop Languages:
# Symbolic Communication for Efficient Multi-Agent Reasoning

**Zhengqi Pei** [1 2]   **Qingming Huang** [1 2]   **Shuhui Wang** [1]

## Abstract

Chain-of-Thought (CoT) improves large language models (LLMs) on difficult reasoning tasks, but it often incurs long natural-language rationales that are poorly aligned with efficient machine reasoning. We propose *Communicative Language Symbolism Routing* (CLSR), a test-time framework in which multiple LLM agents autonomously invent, evolve, and share compact *Language Symbolism Frameworks* (LSFs), while a latent-free router adaptively selects and composes these languages per query to optimize the accuracy–token trade-off. Unlike prompt optimization that refines surface instructions, CLSR treats each LSF as a reusable symbolic protocol with compact symbols, usage rules, and a message-passing contract, and improves it through an evolutionary loop driven by correctness and token cost. At inference time, the router may invoke a single low-cost LSF call, ensemble multiple LSFs, or execute a multi-round LSF composition protocol on harder queries. Across challenging benchmarks, CLSR reduces latency-oriented generated token completion by $3 \sim 6\times$ compared to standard CoT while maintaining accuracy. We further derive an information-theoretic lower bound on token cost under arbitrary symbolism and show that, under an interpreter-realizability premise, multi-round LSF protocols conditionally subsume program-execution pipelines. Code is publicly available[1]

[1]State Key Lab of AI Safety, Institute of Computing Technology, Chinese Academy of Sciences, Beijing, China. [2]School of Computer Science and Technology, University of Chinese Academy of Sciences, Beijing, China.. Correspondence to: Shuhui Wang <wangshuhui@ict.ac.cn>.

*Proceedings of the 43rd International Conference on Machine Learning*, Seoul, South Korea. PMLR 306, 2026. Copyright 2026 by the author(s).

[1]https://github.com/pzqpzq/LSF_MDia

## 1. Introduction

Large Language Models (LLMs) have demonstrated remarkable reasoning abilities, especially when employing chain-of-thought and self-consistency strategies that explicitly generate intermediate reasoning steps (Wei et al., 2022; Yao et al., 2023; Wang et al., 2025a; Besta et al., 2024). Prompting an LLM to produce a step-by-step explanation often improves its accuracy on complex tasks (Zhang et al., 2025b; Sprague et al., 2025). However, these reasoning chains are usually expressed in natural language designed for human understanding. Linguistic reasoning chains can be verbose and are not specifically designed for compact logical inference (Turpin et al., 2023; Tanmay et al., 2025), thus they may not be the most efficient internal representation of the model (Yu et al., 2024; Chen et al., 2025), especially for budget-constrained deployments (Nayab et al., 2024b; Arora & Zanette, 2025). This raises an intriguing question that we aim to address in this study: Can LLMs *invent* new symbolic languages to reason more efficiently?

Recent work leaves large gaps in our question. Prompt optimization methods (Yang et al., 2023; Fernando et al., 2024; Zhou et al., 2024; Aytes et al., 2025b) primarily refine natural-language instructions rather than inducing a persistent compositional symbol system that can be reused and transferred between tasks. Program-based methods (Gao et al., 2023; Chen et al., 2023; Grand et al., 2024) heavily depend on human-designed intermediate languages or programs, therefore, do not study whether LLMs can self-discover discrete symbolic languages optimized for their own tokenization and inductive biases. RL-based approaches (Tanmay et al., 2025; Gehring et al., 2025) produce compact symbolic traces using verifier-based RL, yet these RL pipelines typically incur substantial training cost and optimization instability, making them heavy-weight and less modular for rapid multi-agent language evolution. Compression methods (Cheng & Van Durme, 2024) reduce the length of explicit reasoning by moving computation into dense tokens, but they do not yield a socially shareable discrete "language" that can evolve via inter-agent adoption and merger. Neuro-symbolic approaches (Beiser et al., 2025) and logical reasoning (Xu et al., 2025a) show that an intermediate language can motivate a systematic search over

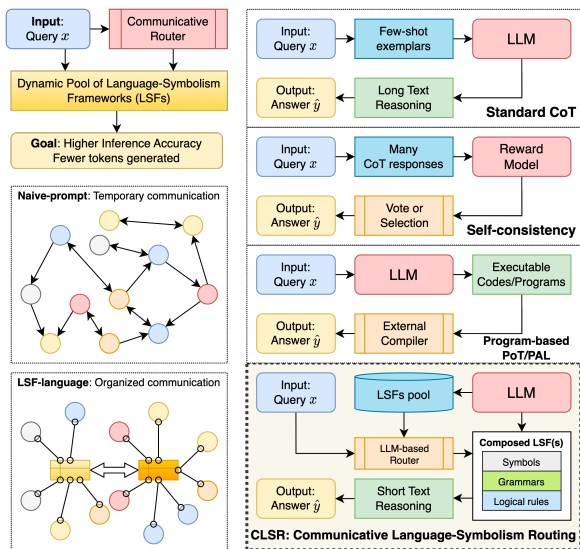

*(a)* Conceptual comparison.

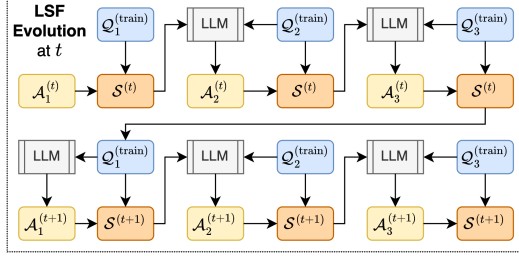

*(b)* Multiple agents evolve new LSFs.

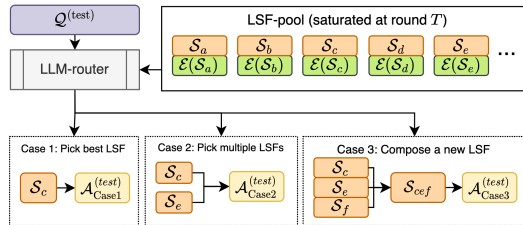

*(c)* Query-adaptive LSF planning.

*Figure 1.* **Communicative Language Symbolism Routing (CLSR). (a)** We let LLMs self-evolve an LLM-oriented compact *Language Symbolism Framework* (LSF), *i.e.*, an information-dense reasoning "dialect", rather than expanding natural language rationales (*e.g.*, CoT) or relying on an external executor (*e.g.*, PoT). An *LLM router* then selects, ensembles, or composes LSFs, enabling a controllable accuracy–token trade-off. **(b)** Multiple agents iteratively propose, critique, and mutate candidate LSFs based on high-leverage selection. **(c)** The saturated LSF pool is then queried at inference time for query-adaptive protocol planning.

representations; yet, their pipelines typically select among fixed formal languages rather than evolve new ones in a socially grounded way.

In parallel, emergent communication methods (Lazaridou & Baroni, 2020; Tucker et al., 2022; Pei et al., 2024a; 2025) formalize how discrete protocols can arise under computational pressures, offering an information-theoretic lens on why structured conventions emerge. Motivated by this perspective, we ask whether LLM agents can autonomously design and evolve languages, then subject them to natural selection. We operationalize this question as an evolutionary search over reusable LSFs: protocols that are simultaneously accurate, compact, and reusable are preferentially retained and made available for downstream routing. This setting yields emergent large-scale symbolic systems shaped by collective use rather than isolated prompt tuning[2].

In this work, we introduce CLSR (Communicative Language Symbolism Routing), a paradigm in which multiple LLM agents autonomously develop their own symbolic communication systems for reasoning. Each agent is a black-box LLM instance that proposes, evaluates, or refines a Language Symbolism Framework (LSF), where an LSF is a customized set of symbols, syntax, and usage rules that serves as a reusable reasoning protocol. We emphasize that LSFs are not manually pre-defined by humans as a fixed formal language. They are induced by the LLMs from exemplars and refined through selection. Thus, our claim is

not that LSFs are free of human priors. Rather, the claim is that the *operational protocol* used at inference time can be automatically generated, selected, shared, and composed by LLM agents, rather than being crafted manually as a fixed prompt or program.

This view naturally connects CLSR to a socio-linguistic perspective inspired by the cultural language evolution process (Kouwenhoven et al., 2025). Across evolutionary rounds, more effective LSFs are preferentially retained and reused, while less accurate or overly verbose LSFs are discarded. The resulting process resembles cultural language evolution under an explicit communication bottleneck: correctness plays the role of communicative success, token length plays the role of production cost, and routing plays the role of pragmatic code-switching between compact and redundant LSFs. Over generations, more effective LSFs spread across agents, while less useful ones are discarded, leading to converging robust symbolic languages.

We design experiments on a diverse set of challenging tasks, including knowledge-intensive QA (Wang et al., 2024; Rein et al., 2023; Lu et al., 2022), mathematical problem solving (Cobbe et al., 2021; Hendrycks et al., 2021; Veeraboina, 2023), and multi-hop reasoning (Yang et al., 2018), to evaluate the impact of LLM-created languages. For generality, we conduct experiments with various open-source LLMs, *e.g.*, Qwen3-8B/32B (Yang et al., 2025), LLaMA3-8B (AI@Meta, 2024), and DeepSeek-R1 (DeepSeek-AI, 2025), treating each model as an agent in different runs.

---

[2]See the Appendix F for more discussion of related work.

Across multiple open-source backbones, CLSR improves the Pareto frontier of accuracy versus generated tokens, outperforming baselines covering token-reduction, program-execution, and prompt optimization.

Ablations further show that (i) deeper evolutionary generations systematically shift languages toward better accuracy-per-token, (ii) scaling the agent population improves the chance of discovering robust, reusable LSFs, provided the selection objective balances correctness and compression. These empirical results lead to an information-theoretic interpretation regarding the curve between reasoning accuracy and token efficiency, given any language symbolism. We also conduct larger-backbone transfer, population-size ablations, cache-aware token accounting, and category-free domain transfer to validate methodological robustness. In summary, this work presents an approach to improving LLM reasoning through socially emergent symbolic communication. Our main contributions include:

- **CLSR: routing over emergent languages.** We propose a new test-time paradigm that treats LLM-created symbolic languages (LSFs) as modular "experts" and routes, ensembles, or composes them per query to explicitly optimize the accuracy–token budget trade-off.
- **LLM-driven language synthesis and evolution.** We introduce an evolutionary bootstrapping procedure that generates LSFs from scratch; we further enable cross-agent adoption and cross-benchmark pooling to promote the transferring and merging of LSFs.
- **Strong empirical Pareto gains.** On seven diverse reasoning benchmarks and multiple open-source backbones, CLSR consistently improves the accuracy–efficiency frontier, outperforming representative token reduction baselines.
- **A theory of accuracy–token optimality under symbolism.** We formalize CLSR as a constrained stochastic control problem, derive an information-theoretic lower bound on minimal token cost, and characterize multi-round multi-LSF protocols as a conditional generalization of program-execution inference.

## 2. Methodology

We introduce the CLSR framework for test-time reasoning compression through LLM-generated symbolic languages. Unlike prompt optimization that searches for instructions or alters surface phrasing, CLSR optimizes *a reusable communication protocol*: (i) the intermediate messages are constrained by grammar/contract and validated; (ii) the same LSF is reused across many queries (not per-instance prompt tuning); (iii) the router explicitly composes multiple LSF calls into a multi-round protocol under a cost budget. During inference, the LLM participates in a multi-round communication where different LSFs act as specialized "experts",

and a router adaptively selects which LSFs should speak, in which order, and when to stop, explicitly optimizing the accuracy–token trade-off. These "experts" are emergent symbolic languages rather than separate neural modules, and the LLM's weights remain inaccessible.

### 2.1. Problem Setup and Metrics

Let $x$ be a query and $y$ be the ground-truth answer drawn from a benchmark. Suppose that we have an evolved LSF pool $\mathcal{S} = \{\mathcal{S}_k\}_{k=1}^K$. In inference, CLSR runs for $T$ rounds, where $T$ can be fixed or determined by the router based on the test query. At round $t \in \{1, \ldots, T\}$, the router selects an index set $\mathcal{I}_t \subseteq [K]$, where $[K] = \{1, \ldots, K\}$, and invokes the corresponding LSF subset $\{\mathcal{S}_k : k \in \mathcal{I}_t\}$. The LLM then produces one response per selected LSF. Empirically, a harder problem often yields a larger $T$ or $|\mathcal{I}_t|$.

**Compute cost (token accounting).** Let $r_{t,k}$ denote the LLM response produced in round $t$ under LSF $\mathcal{S}_k$, and let $r_t^{\text{router}}$ denote the LLM-router's generated planning output in round $t$. We define the generation cost as

$$C = \sum_{t=1}^T \left( |r_t^{\text{router}}| + \sum_{k \in \mathcal{I}_t} |r_{t,k}| \right), \tag{1}$$

where $|\cdot|$ counts generated completion tokens. Thus, all online tokens emitted by the LLM are included, including router outputs, intermediate LSF responses, and aggregation-related responses. This metric is motivated by the standard latency decomposition of LLM inference into a prompt *prefill* phase (time-to-first-token, TTFT) and an auto-regressive *decode* phase (time-per-output-token, TPOT). For reasoning workloads with long completions, the decode term is often the dominant latency component, so completion tokens provide a useful latency-oriented proxy. At the same time, LSF profiles do introduce input-token overhead. In deployment, these cards can be arranged as a fixed canonical prompt prefix and reused across many queries, so the relevant overhead depends on prefix caching and serving infrastructure. For this reason, we additionally report a cache-aware token-equivalent metric in Appendix A.8, while keeping $C$ as the main generated-token metric throughout the paper.

**Objective.** Given a budget $B$, our objective is to maximize expected correctness under the expected realized cost:

$$\max_\pi \ \mathbb{E}[\mathbb{I}\{\hat{y} = y\}] \quad \text{s.t.} \quad \mathbb{E}[C] \leq B,$$

where policy $\pi$ specifies the router, stopping, and aggregation rules. The expectation is over the benchmark distribution and any policy-induced randomness.

### 2.2. Language Symbolism Framework (LSF)

**Conceptual definition.** In idealized form, an LSF can be described as a symbolic communication protocol comprising

three components: (i) symbol naming, namely a compact lexicon; (ii) syntax, namely a compositional grammar; and (iii) constraints, namely well-formedness and usage rules. However, in practice, fully hand-designing these components would collapse the method into prompt engineering. We therefore delegate LSF construction to the LLM itself and use human intervention only to define the high-level optimization goal: to preserve reasoning accuracy while reducing token usage.

**Operational LSF card.** In implementation, each LSF is stored as a compact card $S_k = (\mathcal{V}_k, \mathcal{G}_k, \mathcal{R}_k, \psi_k, \rho_k)$, where $\mathcal{V}_k$ is the symbol inventory, $\mathcal{G}_k$ is a lightweight grammar or output schema, $\mathcal{R}_k$ contains usage rules and validity constraints, $\psi_k$ maps symbols or templates to their intended reasoning operations, and $\rho_k$ is an empirical profile summarizing cost, accuracy, domains, and failure modes. The card is represented textually because the backbone is a language model, but its role is not that of a one-off instruction string. An LSF is reused across many queries, produces messages constrained by a grammar or contract, can be selected or composed by the router, and is evaluated as a persistent protocol whose utility is measured over a population of tasks. This operational distinction is important: prompt optimization searches for better surface instructions, whereas CLSR searches over reusable symbolic communication systems and their routing policies.

**Seed exemplars and LSF synthesis prompt.** Given a benchmark training set, we randomly sample a small set of QA exemplars (typically dozens) and place them in context. We then prompt the LLM to freely invent a symbolic language that aims to reduce token usage while preserving reasoning capability, *e.g.*, design an LSF based on the exemplars in the chat history to minimize the number of tokens while maintaining reasoning capacity. Manual editing, symbol pruning, or grammar correction is not performed in the default pipeline. The exact prompt templates for LSF operations are provided in the Appendix B.

**Diversity and positioning.** We observe that a higher sampling temperature naturally yields a diverse LSFs population, spanning a spectrum from strict LSFs (machine-like, compressed, rigid formats) to soft LSFs (closer to natural language but still symbolically structured). This diversity is crucial for downstream selection and routing, as different symbolic protocols may dominate on different sub-distributions of queries. While the generation loop superficially resembles iterative prompt discovery, the optimized object is not an instruction string but a reusable symbolic language system. LSFs are repeatedly invoked across many queries, come with a compact grammar/contract, and can be selected or composed by the router as a modular protocol. This distinguishes CLSR from prompt evolution / optimization methods (Yang et al., 2023; Fernando et al., 2024).

## 2.3. LSF Evolution: Iterative Bootstrapping

We introduce an evolutionary bootstrapping procedure that progressively refines the LSF pool using only training data and black-box LLM calls.

**Agent population.** In our implementation, an "agent" is an independently sampled black-box LLM instance specified by a backbone, random seed, exemplar subset, and LSF-generation context. Agents are not separate trainable neural modules; they are proposal, critique, and mutation workers used to diversify the LSF population. Different agents may discover different surface LSFs, but the selection objective favors LSFs that repeatedly achieve high correctness with low generated-token cost. This definition also clarifies the agent-count ablation in Appendix A.5.3: increasing the number of agents expands the search over possible symbolic conventions, while the final inference model remains frozen.

**Dataset partition and generational schedule.** We divide the training set into $M$ disjoint groups $\{\mathcal{G}_1, ..., \mathcal{G}_M\}$, each containing exemplars $(x, y)$, then sequentially generate:

1. **Generation**. Use exemplars from $\mathcal{G}_1$ to induce an initial LSF pool $\mathcal{S}^{(1)} = \{\mathcal{S}_k^{(1)}\}_{k=1}^K$.
2. **Evaluation**. For each $\mathcal{S}_k^{(1)}$, query the LLM on the queries of $\mathcal{G}_2$, produce answers $\hat{y}$ and record the token cost of each answer.
3. **Selection and mutation**. From all the answers generated on $\mathcal{G}_2$, select a subset of high-leverage (correct and token-efficient) answers and feed them to the LLM to generate the next-generation LSF pool $\mathcal{S}^{(2)}$.
4. **Repeat**. use $\mathcal{S}^{(t)}$ on $\mathcal{G}_{t+1}$, select high-leverage answers, and synthesize $\mathcal{S}^{(t+1)}$ until the performance in the validation set is saturated.

This procedure is reminiscent of iterative optimization in prompt-evolution systems, but differs in that the feedback signal is used to evolve symbol inventories and constraints rather than only instructions in natural-language.

**High-leverage selection and mutation.** Let an LSF-conditioned answer be a tuple $(\hat{y}, c)$, where $c$ is the generated token count. We prioritize the reasoning traces that are: (i) *correct*, meaning $\hat{y} = y$ under the benchmark's evaluator, and (ii) *token-efficient*, meaning that $c$ is low relative to other correct candidates for the same query. In practice, we implement selection via a Pareto criterion, *i.e.*, accuracy first and then minimal tokens. The selected high-leverage traces, together with the parent LSFs that produced them, are then used as context for the mutation step. The LLM is asked to refine or recombine the parent symbolic protocol so that it preserves the successful inference pattern while removing redundant notation, ambiguous rules, or failure-prone conventions. Thus, mutation operates on the LSF definition rather than merely rewriting an output answer.

**Elitist inheritance across generations.** If certain LSFs consistently generate high-leverage answers, we allow them to survive unchanged into the next generation. In practice, the best performing LSFs $\mathcal{S}_k^{(t)}$ are copied into $\mathcal{S}^{(t+1)}$ without modification, while the remaining population is filled with mutated or recombined variants. This prevents the population from losing strong LSFs due to stochastic mutation, while still allowing exploration.

**LSF Profile: Cost–Accuracy Metadata for Routing.** We maintain an LSF's profile summarizing empirical behavior over training rollouts. For each LSF $\mathcal{S}_k$, we track: (i) Accuracy estimate in evaluated groups, (ii) Token statistics (mean/median, tail behavior), (iii) Reliability indications (variance between query types, failure modes), and (iv) Domain tags inferred from the queries where $\mathcal{S}_k$ excel. These profiles are later consumed by a router to decide which LSF(s) to apply in inference.

## 2.4. Test-time LSF Routing

CLSR moves beyond "choose one prompt" and instead treats LSFs as a pool of symbolic protocols that can be selected, ensembled, and composed per query.

**Latent-free LLM-routing policy** To eliminate additional trainable routing networks and access to the latent states of an LLM or network, we can use an LLM-router by delegating routing decisions to the same LLM. This is achieved by (i) compressing each LSF's "battle" profile into a short description and (ii) prompting the LLM to synthesize a query-specific protocol for leveraging the LSF pool.

**LSF profile summarization.** Offline, we ask the LLM to produce a concise descriptor for each LSF conditioned on its empirical profile: what types of queries it tends to solve well, typical token footprint, and common failure cases.

**Protocol planning.** At test time, given a query $x$ and the set of LSF descriptors, the LLM first decides among three inference modes:

1. **Single LSF direct answer**: select one LSF and generate a final answer.
2. **multi-LSF aggregation**: select multiple LSFs to independently propose answers, then aggregate (e.g., majority vote), similar to self-consistency.
3. **Implicit LSF composition**: for hard queries, the router specifies a multi-step protocol where responses from many LSFs become a context for subsequent rounds.

Concretely, the router produces a plan: (i) the number of rounds, (ii) which LSF(s) to invoke per round, (iii) how to post-process intermediate responses, and (iv) when to stop. See Appendix C for more details.

## 2.5. End-to-End Inference Procedure

Given a test query $x$ and an evolved LSF pool $\mathcal{S}$, we:

1. **Retrieve LSF descriptors** and optionally a small subset of profile statistics.
2. **Router planning**: the LLM outputs a protocol specifying which LSF(s) to use, whether to parallelize, and whether to run multiple rounds.
3. **Execute protocol**: generate intermediate responses as specified. All generated tokens across all rounds and all invoked LSFs are counted toward the total cost.
4. **Aggregate to the final answer** according to the router-generated or self-consistency protocol.

This yields an adaptive reasoning pattern: easy queries often require a single low-cost LSF call ($T$=1), whereas difficult queries may trigger multi-round LSF communication ($T$>1), trading extra tokens for higher reasoning reliability and inference accuracy.

# 3. Theoretical Analysis

This section provides a compact, theory-oriented view of CLSR. The formal proofs are deferred to the Appendix E. We also include a sociolinguistic interpretation of the evolution of LSF, which explains why compact machine-oriented LSFs can emerge from repeated selection under an accuracy–efficiency pressure.

## 3.1. Token-Accuracy Optimality

We formalize CLSR test-time inference as a constrained stochastic control problem. For a fixed test query $x$, an interactive inference policy $\pi$ (routing, stopping, aggregation) induces a multi-round transcript $\mathcal{T}$ (all tokens generated across invoked LSFs/rounds) and a final prediction $\hat{Y}$. We evaluate policies according to expected token cost $J_C(\pi) = \mathbb{E}[|\mathcal{T}|]$ and accuracy $J_A(\pi) = \mathbb{P}[\hat{Y} = Y]$, and define the optimal accuracy within the budget $B$ as

$$A^*(B) = \sup_{\pi:J_C(\pi) \leq B} J_A(\pi). \quad (2)$$

**Theorem 3.1.** *(Existence of an optimal interactive inference policy). For a query $x$. Assume: (i) a finite LSF pool of size $K$; (ii) a finite (or effectively bounded) horizon $T_{max}$ via a stopping action; (iii) the per-step token cost is nonnegative and integrable; (iv) the LLM decoding under each LSF defines a well-posed stochastic kernel over the next tokens conditioned on the current history. Then for every budget $B \geq 0$, the supremum $A^*(B)$ is attained: there exists a possibly randomized policy $\pi_B^*$ such that $J_C(\pi_B^*) \leq B$ and $J_A(\pi_B^*) = A^*(B)$.*

Moreover, for any Lagrange multiplier $\lambda \geq 0$, there exists an optimal deterministic finite-horizon policy $\pi_\lambda^*$ that maxi-

mizes the objective $\mathbb{E}\big[\mathbf{1}\{\hat{Y} = Y\} - \lambda|\mathscr{T}|\big]$; the boundary points of the Pareto frontier $(J_C, J_A)$ can be implemented by some $\pi_\lambda^*$ or a mixture of two adjacent multipliers.

**Theorem 3.2. (Lower bound on expected generated tokens).** *Fix a query $x$ and let $\mathcal{Y}_x$ denote the finite set of effective answers induced by the evaluator for $x$. Consider any interactive CLSR policy $\pi$ whose transcript $\mathscr{T}$ is generated token-by-token until a stopping action, and whose final answer is $\hat{Y} = g(\mathscr{T}, x)$. Suppose $\pi$ achieves target accuracy $\alpha \in (0, 1)$: $\mathbb{P}_\pi\big[\hat{Y} = Y \mid X = x\big] \geq \alpha$. Let $\delta = 1 - \alpha$ and define the required information*

$$I_{\text{req}}(x, \delta) = H(Y \mid X = x) - h_2(\delta) - \delta \log_2(|\mathcal{Y}_x| - 1),$$
$$(3)$$

*where $h_2(\cdot)$ is the binary entropy. Let $\kappa_\theta(x)$ be an upper bound on the conditional information revealed by one* active *generated token under the allowed LSF protocols:*

$$\kappa_\theta(x) = \sup_{\pi, t, h_{<t}} I(Y; Z_t \mid X = x, Z_{<t} = h_{<t}, |\mathscr{T}| \geq t),$$
$$(4)$$

*where the supremum ranges over policies, time steps, and reachable active histories. Then*

$$\mathbb{E}_\pi[|\mathscr{T}| \mid X = x] \geq \frac{\max\{I_{\text{req}}(x, \delta), 0\}}{\kappa_\theta(x)}. \qquad (5)$$

The bound separates three factors. First, a higher target accuracy reduces $\delta$ and increases the Fano term, raising the information that the transcript must convey. Second, harder queries have larger evaluator-induced uncertainty $H(Y \mid X = x)$ or larger effective answer classes $\mathcal{Y}_x$, increasing $I_{\text{req}}$. Third, stronger symbolic protocols increase $\kappa_\theta(x)$ by packing more task-relevant information into each generated token. In this sense, CLSR improves efficiency not by violating an information limit but by changing the representation so that each emitted token carries more useful reasoning state. The active-history definition of $\kappa_\theta(x)$ is important for adaptive stopping: a policy may continue only on difficult histories, so the per-token information rate must be conditionally bounded on the process still being active.

### 3.2. A Sociolinguistic Lens on LSF Evolution

The empirical behavior of LSFs has a useful sociolinguistic interpretation. CLSR does not create language from an empty prior: the base LLM has already internalized human linguistic conventions, mathematical notation, domain-specific abbreviations, and tokenizer-specific regularities. Evolved LSFs are therefore best understood as *machine-oriented languages*: they are constrained by pretrained linguistic priors, but selected under a new objective that rewards correctness per generated token.

This lens explains three recurring patterns. First, successful LSFs follow a principle of least effort: they remove low-information narrative text while preserving symbols that carry reusable reasoning states, such as variable bindings, subgoal markers, verification tags, and short operators. Second, evolution induces conventionalization. A symbol survives not because it is intrinsically meaningful to humans, but because it repeatedly supports reliable inference for the model population. Third, routing implements pragmatic code-switching. For easy queries, a strict and terse LSF often suffices; for ambiguous or difficult queries, the router may select a more redundant LSF, aggregate multiple LSFs, or invoke additional rounds for verification.

Accordingly, CLSR should be viewed as a test-time mechanism for studying how LLM agents invent, select, and reuse compact social conventions under an explicit communication bottleneck, rather than as a claim of human-independent language emergence. Appendix E.1 expands this connection to least-effort and iterated-learning principles.

### 3.3. Universality and Finite Programs

The CLSR protocol class can be viewed as a programmable computation system: the router implements control flow (which "expert language" speaks next and when to stop), each LSF implements an instructive symbolic coding regime, the transcript $\mathscr{T}$ is writable memory, and aggregation is an arbitrary computable readout. Next, we demonstrate that the multi-round multi-LSF interaction is a natural generalization of the "generate-a-program then execute" paradigms. Consider two families of inference procedures:

1. **Program-Execution Pipelines (PE).** A well-formed PE procedure makes finite calls to the base LLM and obtains an LLM-generated transcript $\mathscr{T} \in \Sigma$ (*e.g.*, code/program plus any intermediate strings). Then it applies a deterministic logic-only executor $\text{Exec} : \Sigma \mapsto \mathcal{A}$ that does not access external knowledge.
2. **LSF-only CLSR Protocols.** A policy $\pi$ is a multi-round multi-LSF protocol that can adaptively choose LSFs and stopping rules but does not invoke any external executor. It outputs $\hat{y}$ solely from LLM generations.

**Premise (Interpreter Realizability).** For every deterministic computable executor $\text{Exec} : \Sigma^* \mapsto \mathcal{A}$ in the allowed executor class, there exists an LSF $S_{\text{Exec}}$ such that, for any input string $s \in \Sigma^*$, one LLM call conditioned on $S_{\text{Exec}}$ and $s$ produces $\text{Exec}(s)$ with failure probability at most $\varepsilon_{\text{int}} < 1/2$ and with output length at most $|\text{Exec}(s)| + c_0$ for a constant $c_0$. This is an idealized internal-simulation premise. It is supported in principle by universality results for Transformer-like architectures (Pérez et al., 2019; 2021), but it is not automatically guaranteed for any finite pretrained checkpoint.

**Theorem 3.3. (Conditional subsumption of program execution).** *Under interpreter realizability, consider any program-execution pipeline PE that achieves accuracy of*

*Table 1.* **Comparison with Token-reduction approaches on the accuracy–token trade-off).** Inference accuracy (%) and average completion tokens per problem for Raw CoT (Wei et al., 2022), CCoT (Nayab et al., 2024a), CoD (Xu et al., 2025b), SoT (Aytes et al., 2025a), and CLSR (ours) across seven benchmarks and four backbones (LLaMA3-8B, DeepSeek-R1-Qwen3-8B, Qwen3-8B, and Qwen3-32B). CLSR consistently improves the accuracy–token frontier, typically matching Raw CoT accuracy while reducing token usage by roughly $3/4$, and outperforming length-controlled prompting baselines at comparable budgets.

| LLMs | Methods | MMLU-PRO | | GPQA-MAIN | | GSM8K | | MATH-500 | | AIME21-24 | | SCI-QA | | HOTPOT-QA | |
|---|---|---|---|---|---|---|---|---|---|---|---|---|---|---|---|
| | | ACC | TKN | ACC | TKN | ACC | TKN | ACC | TKN | ACC | TKN | ACC | TKN | ACC | TKN |
| LLaMA3-8B | Raw CoT | 38.6 | 960 | 30.4 | 1209 | 84.9 | 124 | 51.6 | 704 | 6.1 | 1705 | 61.3 | 51 | 45.1 | 32 |
| | CoD | 35.8 | 352 | 26.4 | 386 | 80.5 | **51** | 42.2 | 280 | 4.9 | 1041 | 59.8 | 19 | 40.2 | 24 |
| | CCoT | 37.2 | 370 | 29.3 | 450 | 83.6 | 65 | 45.6 | 315 | 5.8 | 1205 | 60.2 | 22 | 44.3 | 28 |
| | SoT | 36.4 | 340 | 28.2 | 398 | 82.3 | 55 | 45.2 | 289 | 5.3 | 1051 | 61.2 | 19 | **45.8** | 21 |
| | CLSR (OURS) | **38.9** | **320** | **30.8** | **362** | 84.3 | 52 | **48.2** | **257** | **6.2** | **1035** | **61.4** | **18** | 45.5 | **14** |
| DEEPSEEK-R1 -0528 -QWEN3-8B | Raw CoT | 71.1 | 1789 | 73.2 | 5380 | 92.2 | 433 | 91.2 | 3544 | 86.2 | 8190 | 71.3 | 125 | 55.8 | 213 |
| | CoD | 68.3 | 582 | **73.2** | 1892 | 86.5 | 204 | 80.5 | 1842 | 77.4 | 3545 | 68.4 | 37 | 51.2 | 56 |
| | CCoT | 64.5 | 516 | 70.4 | 1605 | 90.3 | 220 | 83.4 | 2032 | 79.6 | 3946 | 70.2 | 43 | 53.0 | 65 |
| | SoT | 69.2 | 458 | 71.4 | 1536 | 91.8 | 205 | 84.1 | 1982 | 81.3 | 3266 | 71.0 | 41 | **54.8** | 59 |
| | CLSR (OURS) | **70.3** | **356** | 73.1 | **1326** | **92.4** | 198 | **88.1** | **1753** | **82.5** | **2979** | **71.5** | **30** | 54.5 | **43** |
| QWEN3-8B | Raw CoT | 60.2 | 276 | 49.1 | 1085 | 90.9 | 243 | 87.2 | 878 | 46.1 | 4234 | 77.8 | 75 | 66.4 | 134 |
| | CoD | 55.6 | 125 | 42.7 | 282 | 85.4 | 100 | 74.5 | 389 | 30.2 | 2610 | 76.2 | 46 | 63.2 | 58 |
| | CCoT | 50.2 | 135 | 41.3 | 245 | 89.2 | 125 | 78.4 | 423 | 36.7 | 2805 | 78.2 | 49 | 65.2 | 70 |
| | SoT | 58.5 | 118 | 45.2 | 228 | 90.3 | 107 | 81.3 | 294 | 38.0 | 2637 | **79.3** | 45 | 65.8 | 63 |
| | CLSR (OURS) | **60.4** | **96** | **47.7** | 228 | **91.2** | **89** | **86.8** | **257** | **43.7** | **2361** | 77.8 | **35** | **66.4** | **48** |
| QWEN3-32B | Raw CoT | 67.5 | 405 | 54.2 | 983 | 91.3 | 216 | 89.3 | 845 | 46.8 | 4572 | 79.2 | 81 | 68.4 | 165 |
| | CoD | 64.2 | 318 | 49.2 | 276 | 82.1 | 105 | 81.5 | 365 | 32.5 | 2912 | 77.0 | 53 | 64.6 | 78 |
| | CCoT | 65.1 | 192 | 50.8 | 312 | 88.2 | 94 | 82.0 | 382 | 37.3 | 3025 | 79.1 | 56 | 65.8 | 81 |
| | SoT | 64.3 | 134 | 51.4 | 247 | 90.5 | 112 | 83.8 | 278 | 38.5 | 2453 | 78.9 | 48 | 66.2 | 72 |
| | CLSR (OURS) | **68.1** | **118** | **54.0** | 209 | **91.6** | **91** | **89.4** | **206** | **45.2** | **2038** | **80.2** | **39** | **68.8** | **55** |

at least $\alpha$ with an expected LLM-generated token cost of $B$. Let $\bar{\ell}_{\text{Exec}}(x)$ denote the expected length of the executor output plus the constant formatting overhead under the PE transcript distribution. Then for any $\delta \in (0, 1)$, there exists an LSF-only CLSR protocol $\pi$ such that

- **Accuracy:** $\mathbb{P}_\pi[\widehat{Y} = Y \mid X = x] \geq \alpha - \delta$;
- **Token cost:** $J_C(\pi) \leq B + O(\bar{\ell}_{\text{Exec}}(x) \log(1/\delta))$.

In short-answer reasoning benchmarks where $\bar{\ell}_{\text{Exec}}(x) = O(1)$, this reduces to $J_C(\pi) \leq B + O(\log(1/\delta))$. In the exact-interpreter and bounded-output regime, the achievable token–accuracy region of CLSR contains that of PE up to an arbitrarily small consensus overhead.

**When the assumption holds or fails.** Interpreter realizability is plausible when: (i) Exec is within the base model's algorithmic competence (e.g., simple arithmetic, bounded symbolic manipulation), (ii) the required execution trace fits within the model's effective context/attention constraints, and (iii) decoding is sufficiently reliable (or can be stabilized by self-consistency). It may fail for executors that require an exact long-horizon computation or strict formal guaranties.

## 4. Experiments

### 4.1. Experimental Setup

**Benchmarks.** We evaluate symbolic-language reasoning on seven widely used reasoning benchmarks that span knowledge-intensive QA, multi-hop retrieval-style QA, and mathematical problem solving: (i) broad-domain factual and professional reasoning (MMLU-Pro (Wang et al., 2024)), (ii) expert-level science QA with strong adversarial difficulty (GPQA (Rein et al., 2024)), (iii) grade-school multi-step arithmetic (GSM8K), (iv) competition-style proof and derivation (MATH500 (Hendrycks et al., 2021) and AIME (21-24) (Veeraboina, 2023)), (v) science QA with short final answers (ScienceQA (Lu et al., 2022)), and (vi) multi-hop question answering (HotpotQA (Yang et al., 2018)).

**Backbone LLM-agents.** To test generality across model families and scales, we run our pipeline with multiple open LLM backbones, treating each checkpoint as an independent agent in language creation and/or inference. We report the results for Qwen3-8B/32B (Yang et al., 2025), LLaMA3-8B (AI@Meta, 2024), and a distilled variant of DeepSeek-R1 (0528-Qwen3-8B) (DeepSeek-AI, 2025). Throughout, the backbone weights remain *frozen*; improvements come purely from test-time symbolic protocols and routing.

**Baselines.** We compare test-time reasoning and token-reduction prompting strategies: (i) Raw CoT (standard chain-of-thought prompting) (Wei et al., 2022), (ii) CoD (Chain-of-Draft) (Xu et al., 2025b), (iii) CCoT (Constrained CoT) (Nayab et al., 2024a), and (iv) SoT (Sketch-of-Thought) (Aytes et al., 2025a). We also compare program-based strategies: (v) PoT (Program-of-Thoughts prompting) (Chen et al., 2023) and (vi) PAL (Program-aided Language) (Gao et al., 2023). Additionally, we compare prompt optimization methods: (vii) Plan-to-Solve (Wang et al.,

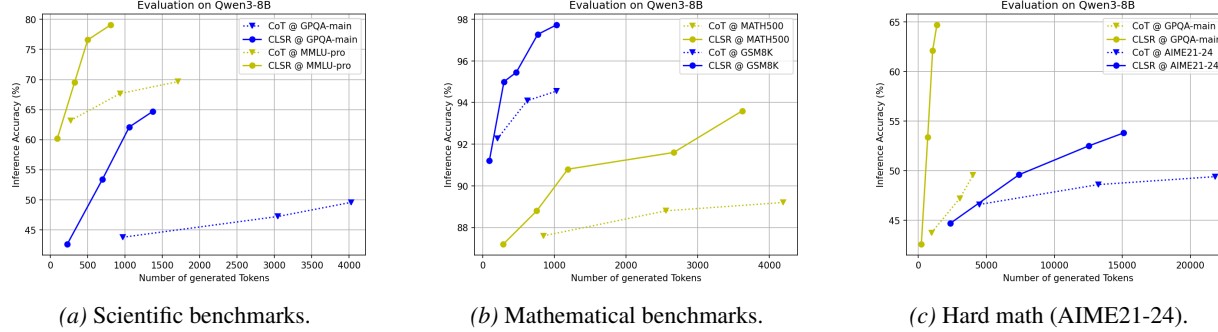

*(a)* Scientific benchmarks.    *(b)* Mathematical benchmarks.    *(c)* Hard math (AIME21-24).

*Figure 2.* **Scaling generated tokens: CLSR dominates CoT under test-time scaling.** We increase the number of generated tokens via test-time scaling (multiple samples with majority vote) and compare standard CoT against CLSR (LSF). The reported curves trace the empirical accuracy–token trade-off. CLSR achieves higher accuracy at a given token budget (or the same accuracy at fewer tokens) across (a) scientific QA benchmarks, (b) math-reasoning benchmarks, and (c) hard math (AIME21–24), supporting our theoretical framing that structured symbolic traces improve the information carried per generated token.

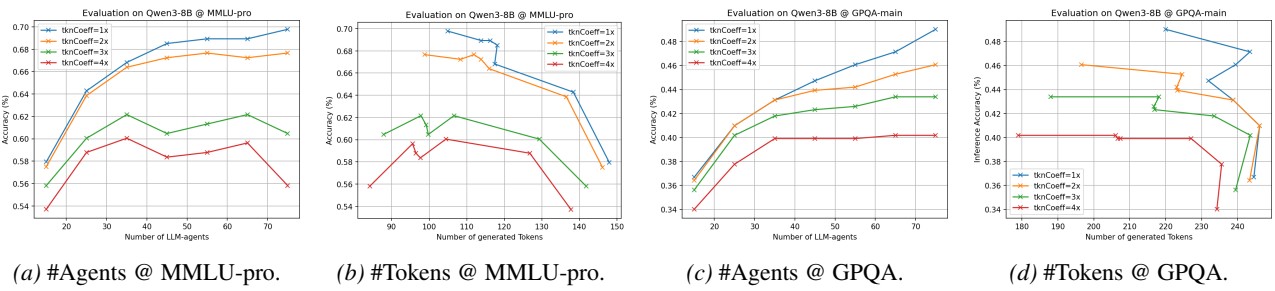

*(a)* #Agents @ MMLU-pro.    *(b)* #Tokens @ MMLU-pro.    *(c)* #Agents @ GPQA.    *(d)* #Tokens @ GPQA.

*Figure 3.* **Scaling evolution agents improves LSF quality.** We ablate the number of LLM agents participating in the offline LSF-evolution process (parallel proposal, critique, and refinement), and evaluate the resulting LSF pools on representative benchmarks. Across tasks, using more agents typically improves downstream accuracy, while also increasing (or modestly changing) generation cost.

2023) and (viii) PromptBreeder (Fernando et al., 2024). All methods are evaluated on the same benchmark splits.

**Metrics (accuracy and token cost).** For each benchmark, we report: (i) inference accuracy (%) and (ii) the number of generated tokens emitted by the inference LLM and the LLM-router. For CLSR, this includes all tokens generated online in selected LSFs, router plans, intermediate rounds, and aggregation responses, which correspond to the cost definition in Section 2.1. We use generated tokens as the main latency-oriented metric because the decode phase dominates many reasoning workloads, whereas prompt-prefix costs are highly serving-dependent. To explicitly address the input-overhead issue, Appendix A.8 reports a cache-aware token-equivalent metric that includes reusable LSF-card prefixes and separates uncached input, cached input, and output-token coefficients.

### 4.2. Implementation Details

**LSF synthesis from exemplars.** For each benchmark, we sample $N \in [200, 2000]$ training exemplars that include ground-truth reasoning content. Given these exemplars in context, we prompt the LLM backbone to invent an initial population of hundreds of Language-Symbolism Frameworks (LSFs), each specifying (i) a compact lexicon, (ii) a

constrained grammar, and (iii) usage constraints that aim to reduce token footprint while preserving reasoning capability. Appendix A.6.4 reports a population-size ablation. We use temperature $0.9$ to generate LSFs and temperature $0.3$ to evaluate the inference.

**Evolutionary bootstrapping (generations).** We iteratively refine the LSF population via an evolutionary loop. At each evolution step, we evaluate candidate LSFs on a held-out validation set, then select high-leverage inference traces that are simultaneously correct and token-efficient. These selected traces are fed back to the LLM to synthesize the next-generation LSF population. We use the fifth evolution LSF pool for the final test evaluation by default.

At inference time, we maintain a global pool by mixing LSFs evolved from different benchmarks. Given a test query, the router may (i) select a single existing LSF, which is typical for easier queries, (ii) select and aggregate multiple LSFs for moderately difficult queries, or (iii) compose multiple existing LSFs across rounds for hard cases. By default, the selection objective weights token length and accuracy equally; the number of tokens of an LSF card ranges from 500–2000 depending on task difficulty and protocol depth. We use the LLM-router to determine the number of rounds $T$ required for the test problem. We can also fix $T=\{1, 3\}$

rather than letting it be determined by the router; see Table 3 in Appendix A for more ablation studies.

**System and compute.** All experiments are conducted on $8\times$ NVIDIA RTX 4090 GPUs. Generating and evolving LSF populations takes roughly several days in total. This is a one-time offline protocol-discovery cost, analogous to the pre-deployment search cost in automatic prompt/protocol optimization methods. CLSR is intended for settings where the evolved LSF pool is reused across many downstream queries, so the offline cost can be amortized over serving-time inference. The online routing/LSF-selection stage takes $< 1$ seconds per query and is small relative to LLM decoding. See Appendix A.8 for details.

### 4.3. Main Results

Tables 1-3 show the central empirical pattern: CLSR consistently improves the accuracy–token frontier across backbones and benchmarks. Compared with Raw CoT, CLSR uses substantially fewer completion tokens while preserving, and in many cases slightly improving, accuracy. Compared with direct token-reduction prompting baselines, our CLSR is more stable on the Pareto frontier: aggressive compression baselines often save tokens by sacrificing correctness, whereas CLSR retains a reusable symbolic protocol and routes among LSFs according to query difficulty.

The gains are not uniform token shortening; they reflect adaptive allocation of reasoning effort. On knowledge-intensive QA benchmarks such as MMLU-Pro and GPQA, much of the natural-language rationale can be compressed into compact symbolic state without a large accuracy loss. On deliberation-heavy mathematical benchmarks such as MATH500 and AIME, the router often benefits from selecting stricter LSFs or invoking additional rounds, because the task requires multi-step transformation and verification. For shorter-answer datasets, the absolute completion length is already small, so the token savings are naturally more limited, but CLSR remains competitive in accuracy.

The figures further explain the mechanism behind the aggregate results. Figure 2 shows that CLSR scales more favorably with additional test-time tokens than standard CoT, indicating that structured symbolic traces use extra budget more efficiently than repeated natural-language sampling. Figure 3 shows that increasing the number of LSF-generating agents improves the final language pool, consistent with broader LSF exploration and stronger selection pressure. Figure 4 shows that deeper evolution improves the accuracy–token ratio before saturation, suggesting that the evolutionary loop is not merely shortening outputs but selecting conventions that preserve more task-relevant information per generated token.

Finally, Table 2 compares CLSR with program-execution

*Table 2.* **Comparison with program-execution and prompt optimization methods on Qwen3-8B.** We compare CLSR with two program-execution pipelines: PoT (Program-of-Thoughts) and PAL (Program-aided Language), and two prompt optimization methods: P2S (Plan-to-Solve) and PBrd (PromptBreeder). For CLSR, $T$ denotes the number of rounds. For PoT and PAL, the reported tokens count only tokens used to generate the program.

| METHODS | GSM8K | | MATH500 | | GPQA | |
|---|---|---|---|---|---|---|
| | ACC | TKN | ACC | TKN | ACC | TKN |
| CoT | 91.6 | 248 | 87.8 | 905 | 43.8 | 973 |
| PoT | 92.1 | 113 | 88.1 | 375 | / | / |
| PAL | 92.5 | 148 | 88.6 | 422 | / | / |
| P2S | 92.2 | 316 | 88.9 | 1205 | 45.2 | 1085 |
| PBRD | 93.6 | 235 | 88.4 | 843 | 46.1 | 922 |
| CLSR | 92.8 | 90 | 86.8 | 257 | 42.6 | 228 |
| CLSR ($T{=}1$) | 92.1 | 83 | 86.2 | 134 | 40.9 | 182 |
| CLSR ($T{=}3$) | **94.8** | 214 | **89.7** | 417 | **49.2** | 565 |

and prompt-optimization baselines under matched settings. CLSR remains competitive without relying on an external executor, and it also improves upon prompt-optimization baselines. These results support the Section 3: CLSR is not a fixed brevity heuristic, but an adaptive policy over the accuracy–token frontier whose multi-round LSF protocols can approximate program-like computation when the required symbolic operations remain within the model's internal competence.

### 4.4. Ablations and Analysis

We ablate the following design choices: (i) evolution depth (Sec. A.5.1), (ii) exemplar count (Sec. A.5.2), (iii) agent count and length-coefficient (Sec. A.5.3), (iv) number of multi-rounds $T$ (Sec. A.5.4), (v) cross-LLM transfer via swapping the LSF generator (Sec. A.5.5), and (vi) qualitative examples (Sec. A.7). We also report robustness studies in Appendix A.6, including seed stability, population-size sensitivity, larger-model inference, long-context pilot evaluation, category/full-pool routing, cross-domain transfer, and cache-aware token accounting.

## 5. Conclusion and Limitations

We introduced CLSR, a test-time framework that lets LLMs invent, evolve, and route among compact symbolic communication protocols. Empirically, CLSR improves the accuracy–efficiency frontier across diverse reasoning benchmarks, with robustness checks showing stable gains. Conceptually, CLSR suggests that LLMs can develop machine-oriented languages that are not manually crafted by humans. However, the offline language-evolution can be computationally expensive, and the synthesis pipeline requires training exemplars with reasoning content. Future work should study how symbolic languages transfer across modalities, tools, and multi-agent environments.

## Acknowledgements

This work was supported in part by the National Key R&D Program of China under Grant 2023YFC2508704, in part by the National Natural Science Foundation of China under grant number 62236008, in part by the Natural Science Foundation of Beijing under grant number L251082, and in part by Shandong Provincial Natural Science Foundation under project ZR2025ZD01.

## Impact Statement

This work aims to improve the efficiency of LLM reasoning by reducing unnecessary generated tokens while preserving task performance. The main potential risk is that compact symbolic traces may be less directly interpretable than natural-language rationales, which could make human auditing more difficult in real-world deployments. A practical mitigation is to use CLSR with larger LLMs for internal reasoning while requiring the final system to output a human-readable explanation using smaller LLMs. We do not introduce new datasets containing personally identifiable information or new model-training procedures beyond black-box LLM prompting and evaluation.

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

# A. Appendix: more empirical results

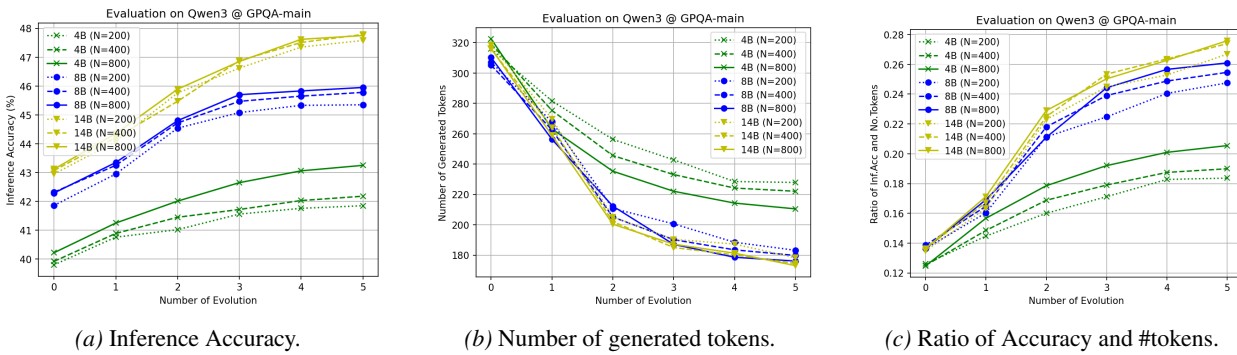

*(a)* Inference Accuracy.  *(b)* Number of generated tokens.  *(c)* Ratio of Accuracy and #tokens.

*Figure 4.* **Effect of evolution depth on the accuracy–token frontier.** We vary the number of language-evolution iterations used to refine the LSF pool, while holding the backbone fixed. $N$ denotes the number of training exemplars used for LSF synthesis. We report (a) test accuracy, (b) average generated completion tokens per problem, and (c) an aggregated efficiency score. Increasing the evolution depth improves both accuracy and token efficiency and then saturates, consistent with the view that better-evolved LSFs increase the effective per-token information rate predicted by Theorem 3.2.

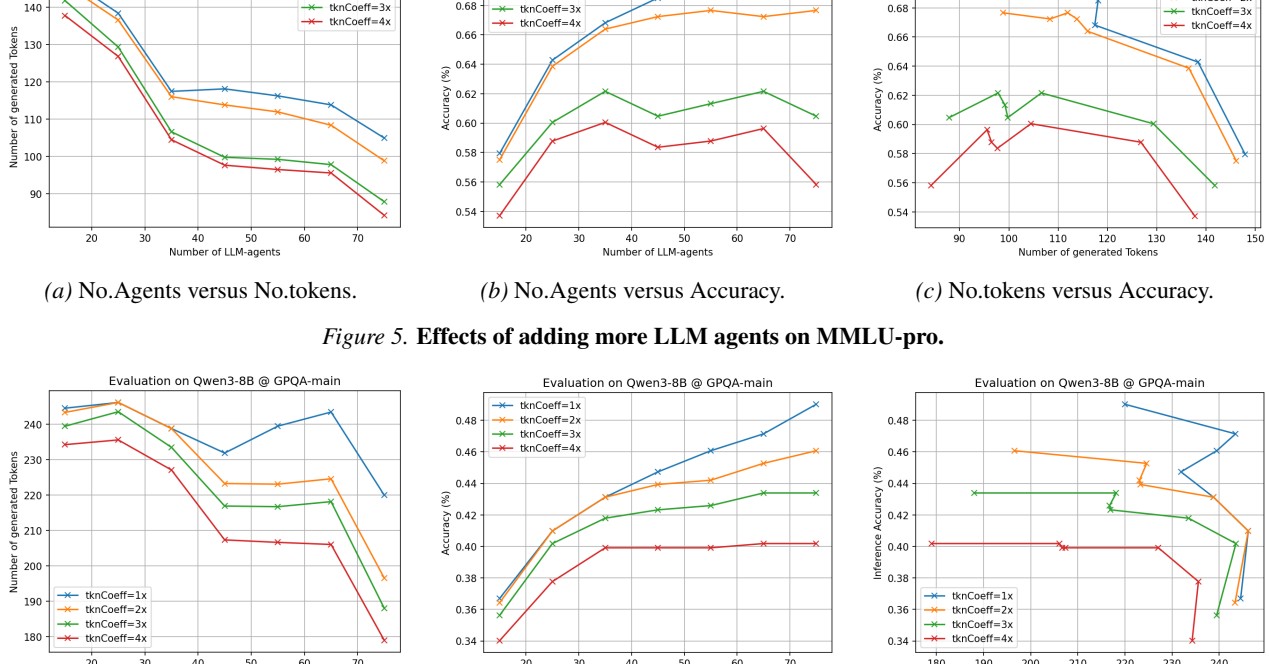

*(a)* No.Agents versus No.tokens.  *(b)* No.Agents versus Accuracy.  *(c)* No.tokens versus Accuracy.

*Figure 5.* **Effects of adding more LLM agents on MMLU-pro.**

*(a)* No.Agents versus No.tokens.  *(b)* No.Agents versus Accuracy.  *(c)* No.tokens versus Accuracy.

*Figure 6.* **Effects of adding more LLM agents on GPQA-main.**

## A.1. Comparison to length-controlled prompting baselines.

Compared with Constrained CoT (CCoT), Chain-of-Draft (CoD), and Sketch-of-Thought (SoT), CLSR achieves a stronger overall trade-off: (i) relative to CCoT, CLSR improves accuracy while also using fewer tokens (since CCoT removes verbosity but does not introduce a task-adaptive symbolic code); (ii) relative to CoD, CLSR substantially improves accuracy at a modest token increase, indicating that extreme compression alone is not sufficient, *i.e.*, the intermediate representation must remain *information-dense and compositional*; (iii) relative to SoT, CLSR achieves higher accuracy with a similar token budget, suggesting that our *evolutionary search over LSFs* discovers more task-specialized symbolic operators than a fixed handcrafted constraint family.

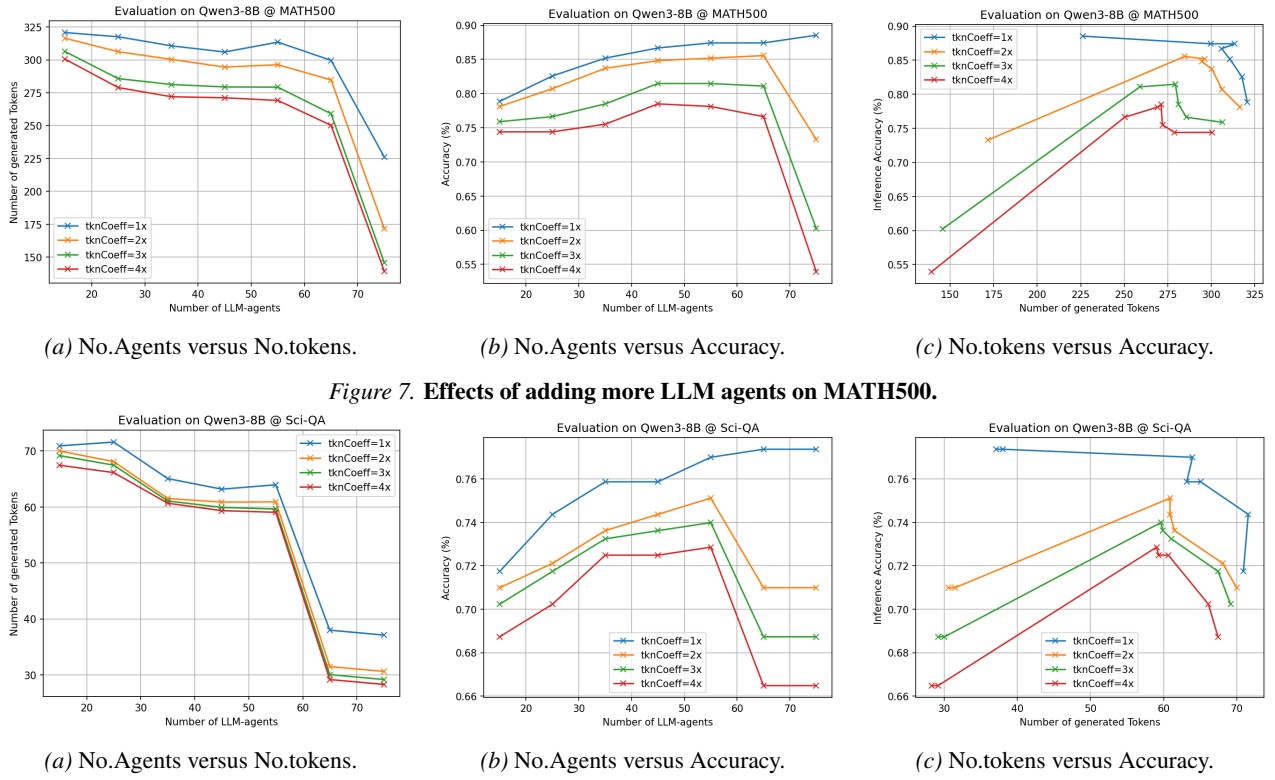

(a) No.Agents versus No.tokens.  (b) No.Agents versus Accuracy.  (c) No.tokens versus Accuracy.

*Figure 7.* **Effects of adding more LLM agents on MATH500.**

(a) No.Agents versus No.tokens.  (b) No.Agents versus Accuracy.  (c) No.tokens versus Accuracy.

*Figure 8.* **Effects of adding more LLM agents on Science-QA.**

### A.2. Interpretation through the theory lens.

Section 3 predicts that, for a fixed target accuracy, token cost is lower-bounded by a ratio $\mathbb{E}[|T|] \geq I_{\text{req}}(x, \delta)/\kappa_\theta(x)$. CLSR improves the empirical cost–accuracy curve in a manner consistent with increasing $\kappa_\theta(x)$: LSF traces replace low-information narrative text with compact, state-carrying symbols (operators, bindings, and verification tags) that better preserve the *computable* parts of reasoning. Moreover, the robust token savings across backbones indicate that CLSR is not merely exploiting idiosyncrasies of a particular model but is instead leveraging a representation-level advantage.

### A.3. Gains are most pronounced on deliberation-heavy problems.

The largest accuracy advantages emerge on benchmarks that require multi-step reasoning and/or verification (notably MATH500 and AIME21–24). For example, CLSR improves AIME21–24 accuracy consistently across backbones while reducing tokens drastically relative to Raw CoT. This pattern aligns with the lower-bound perspective of Theorem 3.2: as task difficulty increases (higher effective uncertainty in the target solution), the required information $I_{\text{req}}(x, \delta)$ rises, making naive verbosity expensive; CLSR responds by increasing the effective per-token information rate $\kappa_\theta(x)$ via a structured symbolic alphabet and reusable operators, thereby achieving the same (or higher) accuracy with fewer tokens.

### A.4. Comparison with program-execution pipelines.

Table 2 compares CLSR with two canonical *program* baselines, *e.g.*, PoT (Chen et al., 2023) and PAL (Gao et al., 2023), which prompt the LLM to emit executable code and delegate the final computation to an external interpreter/runtime. Both PoT and PAL are known to improve arithmetic/symbolic reliability by offloading computation outside the LLM, *i.e.*, "generate a program then execute". On Qwen3-8B, PoT/PAL indeed outperform standard CoT on GSM8K and MATH500 at substantially lower *LLM decoding* tokens. However, CLSR can match or exceed these gains *without any external executor*: with a small majority-vote ensemble ($T = 3$), CLSR reaches 94.8% on GSM8K and 89.7% on MATH500 using 214 and 417 tokens, respectively, exceeding both PoT and PAL in accuracy–token frontier. Importantly, Table 2 reports PoT/PAL token counts only for *program generation*; their end-to-end deployment cost also includes external execution time and tool-chain overhead. From the theoretical lens in Section 3, these results empirically support our claim: under interpreter realizability, CLSR's multi-round multi-LSF protocols can internally emulate the effect of a deterministic executor (Theorem 3.3). In

practice, CLSR provides a *purely in-model* alternative to PoT/PAL that preserves their accuracy benefits while avoiding dependency on external runtimes and execution-layer engineering.

## A.5. Ablation studies

### A.5.1. EFFECT OF EVOLUTION DEPTH.

As shown in Fig. 4, we ablate the *evolution depth*, *i.e.*, the number of recursive propose→critique→mutate→select rounds used to build the LSF pool offline. Increasing the depth consistently improves downstream accuracy at a fixed token budget, indicating that iterative refinement produces more *compressive* and *task-aligned* symbolic protocols rather than merely longer rationales. Practically, the gain tends to saturate after a small number of rounds: once the pool contains a few high-leverage LSF "families", additional rounds mostly perform local polishing (marginal accuracy improvements) while increasing offline cost. Overall, deeper evolution strengthens the LSF pool by amplifying selection pressure toward high-information tokens, but exhibits diminishing returns beyond a moderate depth. This also supports the core hypothesis of *cultural selection* among LSFs: iteratively retaining high-leverage traces (correct & concise) biases the language pool toward more compressive, reusable abstractions.

### A.5.2. EFFECT OF EXEMPLAR COUNT.

As shown in Fig. 4, we vary the *generation batch size*, *i.e.*, the number of candidate LSFs proposed per generation, to characterize the exploration–efficiency trade-off in offline search. A larger batch improves diversity and reduces the probability that evolution prematurely converges to a suboptimal symbolic convention, which in turn increases the chance that the final pool contains a strong specialist for each query type. For larger backbones (*e.g.*, moving from 4B to 14B), increasing $N$ yields diminishing marginal returns in both accuracy and token reductions, suggesting that higher-capacity models can infer a useful symbolic protocol from fewer examples, while smaller models benefit more from larger set of exemplars. In general, batch size mainly controls *exploration*; use it to avoid early collapse but expect diminishing returns once coverage of distinct LSF modes is achieved.

### A.5.3. EFFECT OF AGENTS COUNT AND LENGTH REGULARIZATION.

As presented in Figure 3 and Appendix Figures 5–8, we study (i) the number of agents participating in evolution and (ii) the *length-coefficient* that penalizes verbose intermediate representations during selection. More agents typically improve the pool because it increases stylistic and algorithmic diversity, enabling stronger cross-agent critique and yielding more robust LSFs that generalize across query distributions. Meanwhile, the length-coefficient directly tunes the accuracy–token Pareto point: increasing it encourages shorter, higher-density symbolic traces (fewer tokens), but can under-allocate "reasoning bandwidth" on hard instances if set too aggressively. Overall, agent scaling primarily boosts *search breadth and critique quality*, whereas the length-coefficient provides a clean knob for *token efficiency* that should be set moderately to avoid sacrificing hard-case accuracy.

### A.5.4. A FIXED NUMBER OF LSF-ROUNDS $T$ OR NOT

CLSR applies the same latent-symbolic refinement operator for multiple rounds. A natural design choice is whether the number of rounds $T$ should be *fixed* for all inputs, or *adaptive* (instance-dependent). From the perspective of our theoretical analysis (Section 3), $T$ acts as an inference-time compute budget: each additional round consumes extra generated tokens, but can reduce the residual error by enabling further compression–verification-correction cycles. The optimal policy is therefore generally *not* a single global $T$, but an *adaptive stopping rule* that continues refinement only when the expected marginal gain outweighs the marginal token cost.

Formally, let $x$ be the input, $y$ the ground-truth answer, and let $s_t$ denote the internal state of the CLSR after round $t$ (e.g., the current latent symbolic form and its distribution of induced answers). Each round incurs an expected token cost $c(s_t)$ and yields an expected utility improvement $\Delta u(s_t)$. A standard Lagrangian view of the accuracy–token trade-off is

$$\max_{\pi} \ \mathbb{E}\big[u(\hat{y}, y)\big] \ - \ \lambda \mathbb{E}\Big[ \sum_{t=1}^{T(\pi, x)} c(s_t) \Big], \tag{6}$$

where $\pi$ is a stopping policy deciding whether to continue at each $s_t$. This objective implies an "optimal" behavior that is instance-adaptive: easy problems should stop early (small $T$) to avoid wasting tokens, whereas hard problems may require

additional rounds.

*Table 3.* **Varying the number of CLSR rounds** $T$ **on Qwen3-8B.** We compare standard CoT with CLSR under (i) *fixed* $T = 1$ (single refinement), (ii) *fixed* $T = 3$ (always run three rounds), and (iii) *unfixed* $T$ with $T_{\max} = 3$ (a learned instance-adaptive stopping rule). Accuracy (%) is reported on GSM8K, MATH500, GPQA-main, AIME (2021-2024), and MMLU-pro, along with the average completion tokens per problem. Adaptive CLSR attains the best accuracy with only a modest token overhead relative to fixed-$T$ settings, and it stops early on most instances, demonstrating effective instance-wise compute allocation.

|  | GSM8K | | MATH500 | | GPQA | | AIME | | MMLU | |
|---|---|---|---|---|---|---|---|---|---|---|
|  | ACC | TKN | ACC | TKN | ACC | TKN | ACC | TKN | ACC | TKN |
| CoT | 91.6 | 248 | 87.8 | 905 | 43.8 | 973 | 46.6 | 4502 | 63.2 | 272 |
| CLSR (DEFAULT) | 92.8 | 90 | 86.8 | 257 | 42.6 | 228 | 44.7 | 2361 | 63.6 | 93 |
| CLSR ($T$=1) | 92.1 | 83 | 86.2 | 134 | 40.9 | 182 | 38.4 | 1047 | 62.1 | 85 |
| CLSR ($T$=3) | **94.8** | 214 | **89.7** | 417 | **49.2** | 565 | **46.8** | 3314 | **67.2** | 257 |

Empirically, Table 3 validates this prediction in Qwen3-8B. Compared with $N$-shot CoT, CLSR improves both accuracy and token efficiency. More importantly, *unfixed* CLSR (adaptive $T$, with $T_{\max} = 3$) achieves the best overall accuracy across MATH500 and AIME benchmarks, while keeping the average completion length close to fixed-$T$ settings. In the same experiment, the adaptive policy chooses $T = 1$ for a majority of cases, indicating that CLSR indeed learns to *stop early* when additional refinement is unnecessary, but retains the option to allocate more rounds to difficult inputs. We also observe that for a harder benchmark (*e.g.*, AIME), the default CLSR (unfixed $T$, the LLM-router can determine the value of $T$ based on the test problem) uses fewer cases of $T = 1$, since its corresponding $T = 1$ has fewer tokens compared to the default setting. In general, these results support the core claim of Section 3: CLSR is best viewed as an *adaptive compute allocation* mechanism, rather than a fixed-length prompting recipe.

A.5.5. EFFECT OF SWAPPING THE LSF GENERATOR (CROSS-MODEL LSF TRANSFER)

Table 4 studies cross-model transfer of induced LSF: we evolve the LSF pool using a generator backbone $\mathcal{F}_{\text{gen}}$ but execute CLSR using another inference backbone $\mathcal{F}_{\text{inf}}$. Across GPQA, MATH500, and GSM8K, the matched diagonal ($\mathcal{F}_{\text{gen}}=\mathcal{F}_{\text{inf}}$) is consistently the best in accuracy, while swapping typically reduces tokens but degrades accuracy. Notably, using a stronger generator with a weaker inference model reduces tokens with only mild drops, whereas using a weaker generator with a stronger inference model incurs a larger accuracy penalty; the sensitivity is most pronounced on the knowledge-intensive GPQA benchmark.

*Table 4.* **Swapping the LSF generator (cross-model LSF transfer).** For each inference LLM, we evolve the LSF pool using the specified generator and then run CLSR inference with the inference model. We report inference accuracy (Acc) and average completion tokens (Tkn) on GPQA, MATH500, and GSM8K (tokens include the full executed CLSR protocol). The matched diagonal yields the best accuracy, while swapping generally reduces tokens but degrades accuracy; the degradation is most pronounced on GPQA and is especially severe when the generator is weaker than the inference model.

| INFERENCE LLM | LSF GENERATOR | GPQA | | MATH500 | | GSM8K | |
|---|---|---|---|---|---|---|---|
|  |  | ACC | TKN | ACC | TKN | ACC | TKN |
| QWEN3-4B | QWEN3-4B (DEFAULT) | 45.1 | 329 | 83.2 | 273 | 90.3 | 93 |
|  | QWEN3-8B | 43.5 | 254 | 82.6 | 195 | 90.1 | 86 |
| QWEN3-8B | QWEN3-8B (DEFAULT) | 47.7 | 228 | 86.8 | 257 | 91.2 | 89 |
|  | QWEN3-4B | 42.1 | 172 | 81.8 | 183 | 89.2 | 84 |

This pattern aligns with our theory view in Section 3: CLSR gains come from allocating a limited token budget to a symbolic protocol whose intermediate symbols must be *reliably interpretable* by the inference model. Swapping $\mathcal{F}_{\text{gen}}$ changes the induced codebook (operators/abbreviations/constraints), which can increase a "dialect mismatch" between the protocol and the decoder implemented by $\mathcal{F}_{\text{inf}}$; the result is lower token usage but reduced effective information per token, hence lower accuracy. Practically, the results suggest evolving LSFs with the same backbone as inference when accuracy is the priority, while cross-model transfer is feasible but generally Pareto-suboptimal, especially when the generator is weaker than the inference model.

*Table 5.* **Seed stability.** Accuracy and generated tokens are reported as Acc. (No. tokens). CLSR reports mean $\pm$ std over five random seeds and exemplar samplings.

| BACKBONE | METHOD | MMLU-PRO | GPQA-MAIN | MATH500 |
|---|---|---|---|---|
| LLAMA3-8B | CoT | 38.6 (960) | 30.4 (1209) | 51.6 (704) |
| | SoT | 36.4 (340) | 28.2 (398) | 45.2 (289) |
| | CLSR | $39.2 \pm 0.5\ (316 \pm 13)$ | $30.8 \pm 0.2\ (352 \pm 18)$ | $48.8 \pm 0.7\ (262 \pm 11)$ |
| QWEN3-8B | CoT | 60.2 (276) | 49.1 (1085) | 87.2 (878) |
| | SoT | 58.5 (118) | 45.2 (228) | 81.3 (294) |
| | CLSR | $60.6 \pm 0.4\ (98 \pm 14)$ | $47.9 \pm 0.4\ (216 \pm 12)$ | $86.8 \pm 0.6\ (247 \pm 10)$ |

*Table 6.* **Larger-backbone inference.** Accuracy and generated tokens are reported as Acc. (No. tokens). Results are averaged over three runs.

| BACKBONE | METHOD | MMLU-PRO | GPQA-MAIN | MATH500 |
|---|---|---|---|---|
| QWEN3-32B | CoT | 67.5 (405) | 54.2 (982) | 89.3 (845) |
| | SoT | 64.3 (134) | 51.4 (247) | 83.8 (278) |
| | CLSR | **68.1** (**90**) | 54.0 (**209**) | **89.4** (**206**) |
| LLAMA3.1-70B | CoT | 49.2 (1020) | 41.3 (1358) | 64.3 (725) |
| | SoT | 47.5 (385) | 38.4 (425) | 59.8 (314) |
| | CLSR | **49.8** (**312**) | **42.7** (**354**) | 63.4 (**242**) |

## A.6. Robustness Checks

This section reports additional robustness checks to clarify stability, scale transfer, routing dependence, and practical token accounting.

### A.6.1. SEED STABILITY AND STOCHASTIC MUTATION

Table 5 reports mean $\pm$ std over five random seeds and exemplar samplings. Different runs discover different surface LSFs, but the accuracy–token frontier remains stable.

### A.6.2. LARGER INFERENCE BACKBONES

Table 6 evaluates larger inference backbones using LSFs discovered by Qwen3-8B. The gains persist even when the LSF generator is smaller than the inference model, suggesting that LSFs can transfer as reusable symbolic protocols rather than being tied to one specific checkpoint.

### A.6.3. LONG-CONTEXT PILOT

We also run a small pilot on LongBench-v2 to test whether symbolic protocols remain useful when the input context is long. The results in Table 7 are encouraging, but we treat this as preliminary evidence only. Long-context search agents and repository-level coding require more specialized evaluation and are left for future work.

### A.6.4. POPULATION-SIZE SENSITIVITY

The default initial population size is 100 LSFs. Table 8 shows that this is an empirical exploration/compute trade-off. A very small population lacks LSF diversity, while larger populations yield diminishing returns and higher cold-start cost.

### A.6.5. CATEGORY ROUTING VERSUS FULL-POOL ROUTING

Category routing is a lightweight guiding mechanism rather than a hard partition of capability. Table 9 compares category-routed CLSR with full-pool routing, where the router can select from the entire LSF bank without a pre-specified category filter. Full-pool routing preserves the qualitative gain, indicating that CLSR does not depend on a rigid manually defined taxonomy.

### A.6.6. CROSS-DOMAIN LSF TRANSFER

Table 10 compares in-domain, all-domain, and leave-one-domain-out LSF banks. The results indicate that CLSR benefits are not tied to a single domain-specific LSF. All-domain pooling can slightly improve robustness, while leave-one-domain-out

*Table 7.* **Long-context pilot.** Overall score and average generated tokens are reported.

| BACKBONE | METHOD | OVERALL | AVG. TOKENS |
|---|---|---|---|
| LLAMA3-8B | CoT | 30.8 | 1068 |
| | CLSR | **31.5** | **568** |
| QWEN3-8B | CoT | 38.2 | 842 |
| | CLSR | **39.1** | **432** |

*Table 8.* **Initial LSF population size.** Accuracy and generated tokens are reported as Acc. (No. tokens).

| BACKBONE | INITIAL LSFS | MMLU-PRO | GPQA-MAIN | MATH500 |
|---|---|---|---|---|
| LLAMA3-8B | 10 | 38.1 (354) | 30.2 (425) | 47.4 (312) |
| | 50 | 38.6 (330) | 30.4 (380) | 48.0 (267) |
| | 100 | 38.9 (320) | 30.8 (362) | 48.2 (257) |
| | 200 | **39.2 (308)** | **30.9 (358)** | **48.2 (255)** |
| QWEN3-8B | 10 | 59.2 (103) | 46.6 (234) | 85.4 (275) |
| | 50 | 60.1 (102) | 47.3 (225) | 86.4 (260) |
| | 100 | 60.4 (96) | 47.7 (228) | **86.8** (257) |
| | 200 | **60.6 (93)** | **47.8** (232) | 86.6 (**248**) |

routing still preserves most gains.

## A.7. Qualitative examples: what changes in the trace.

Appendix Figures 9–14 provide representative outputs of the GPQA, MATH500, and AIME.benchmarks. Compared to verbose CoT, our CLSR (LSF) traces replace long natural-language rationales with compact symbolic operators and structured templates that preserve key intermediate state (*e.g.*, variable bindings, sub-goal markers, and verification tags) while discarding redundant narration. We also show the auto-selected LSFs specification used for each query, illustrating that (i) the router selects different LSFs by domain and difficulty, and (ii) the selected LSFs are reusable artifacts rather than one-off prompt rewrites.

This section provides qualitative evidence for *how* CLSR modifies the reasoning trace across refinement rounds. Figures 9–14 show representative model outputs for CoT and CLSR as well as the intermediate LSF produced by CLSR. We highlight several recurring phenomena.

**From narrative CoT to task-relevant *latent symbolic form*.** A consistent difference is that CoT tends to produce verbose narrative intermediate text that mixes (i) problem understanding, (ii) bookkeeping, and (iii) arithmetic/logical execution. In contrast, CLSR compresses the intermediate reasoning into a compact LSF that is closer to a *minimal sufficient representation* to solve the instance: it preserves variable bindings, constraints, and key transformation steps while discarding stylistic and redundant natural-language content. This aligns with the goal of CLSR: to shift reasoning toward a higher signal-to-token ratio without relying on an external executor.

**Error localization and correction across rounds.** Across examples, later rounds frequently correct earlier-round failure modes that are common in plain CoT: (i) missed constraints (*e.g.*, forgetting a boundary condition or a unit conversion), (ii) inconsistent variable definitions, (iii) arithmetic slips that propagate in long traces, and (iv) premature commitment to an incorrect intermediate value. The intermediate LSF makes such issues easier to localize: the model can explicitly re-check constraint satisfaction or recompute a small symbolic sub-part, instead of re-generating a long narrative chain.

**Refinement is often *selective* rather than *additive*.** Importantly, CLSR does not merely append more tokens. Later-round traces often *replace* earlier representations with shorter, cleaner ones: irrelevant branches are removed, equivalent expressions are simplified, and the final answer emerges after a small number of symbolic edits. This behavior explains why increasing $T$ does not necessarily inflate completion length proportionally and why an adaptive policy can stop at $T = 1$ for many inputs.

**When additional rounds help.** The examples suggest that multiple rounds are most beneficial for problems with (i) multi-constraint coupling (where one mistaken assumption breaks downstream steps), (ii) long-range dependencies (where early variable choices constrain later operations), or (iii) heavy arithmetic/combinatorial bookkeeping. In these cases, CLSR

*Table 9.* **Category routing vs. full-pool routing.** Accuracy and generated tokens are reported as Acc. (No. tokens).

| BACKBONE | BANK TYPE | MMLU-PRO | GPQA-MAIN | MATH500 |
|---|---|---|---|---|
| LLAMA3-8B | CATEGORY | 38.9 (320) | 30.8 (362) | 48.2 (257) |
| | FULL-POOL | 38.6 (325) | 30.5 (364) | 48.2 (263) |
| QWEN3-8B | CATEGORY | 60.4 (96) | 47.7 (228) | 86.8 (257) |
| | FULL-POOL | 60.1 (98) | 47.5 (233) | 86.4 (264) |

*Table 10.* **Cross-domain LSF transfer.** Accuracy and generated tokens are reported as Acc. (No. tokens).

| BACKBONE | BANK TYPE | MMLU-PRO | GPQA-MAIN | MATH500 |
|---|---|---|---|---|
| LLAMA3-8B | IN-DOMAIN | 38.7 (312) | 30.6 (358) | **48.4 (252)** |
| | ALL-DOMAIN | **38.9** (320) | **30.8** (362) | 48.2 (257) |
| | LEAVE-ONE-DOMAIN-OUT | 38.0 (345) | 30.3 (393) | 47.9 (268) |
| QWEN3-8B | IN-DOMAIN | **60.5 (92)** | 47.5 (**221**) | 86.8 (**253**) |
| | ALL-DOMAIN | 60.4 (96) | **47.7** (228) | **86.8** (257) |
| | LEAVE-ONE-DOMAIN-OUT | 60.0 (105) | 46.8 (245) | 86.4 (262) |

uses additional rounds to re-encode the problem into a more stable LSF and derive the answer with fewer opportunities for cascading errors.

**Limitations of qualitative traces.** Finally, we emphasize that these traces are presented as *algorithmic artifacts* of CLSR, *i.e.*, intermediate representations used to improve prediction under a token budget, not as guaranteed faithful explanations of internal model causality. Nevertheless, they provide an interpretable window into how CLSR reallocates tokens from verbose narration to structured, constraint-preserving symbolic computation.

### A.8. Latency decomposition and token accounting

**Reference calibration (TTFT/TPOT).** Regarding TTFT (time-to-first-token) and TPOT (time-per-output-token), Table 11 reports representative single-request latency components for Qwen3-8B on RTX 4090 GPU. We use these measurements as a sanity-check that, for long-form reasoning outputs, decoding latency scales approximately linearly with the number of generated tokens and typically dominates prompt prefill overhead.

*Table 11.* **Representative latency decomposition for Qwen3-8B inference (single request, batch size 1).** $L_{\text{in}}$ and $L_{\text{out}}$ denote prompt and completion lengths. TTFT primarily reflects prompt prefill plus the first decode step; TPOT reflects steady-state decoding; RTI denotes the ratio of total latency over token generation (decoding) time.

| $L_{\text{IN}}$ | $L_{\text{OUT}}$ | TTFT (MS) | TPOT (MS/TKN) | RTI |
|---|---|---|---|---|
| 512 | 64 | 76 | 11.2 | 1.106 |
| | 256 | | | 1.026 |
| | 1024 | | | 1.007 |
| 1024 | 64 | 148 | 11.9 | 1.194 |
| | 256 | | | 1.048 |
| | 1024 | | | 1.012 |
| 2048 | 64 | 281 | 12.9 | 1.340 |
| | 256 | | | 1.085 |
| | 1024 | | | 1.021 |

The LLM-router often takes dozens of input tokens to determine the LSF-categories, then takes hundreds of input tokens to read the LSF-profiles and determine which LSFs should be selected; reading a complete LSF often consumes hundreds of tokens. Empirically, the total input token $L_{\text{in}}$ during inference for different benchmarks often ranges from $0.5 \sim 2 \times 10^3$.

**Cache-aware token-equivalent accounting.** The main paper reports generated completion tokens because they are a useful latency-oriented proxy for reasoning workloads. However, an LSF card also contributes input tokens. In a serving system with prompt-prefix caching, a fixed bank of LSF cards can be arranged in canonical order as a reusable prefix, while only the query, compact routing suffix, and generated outputs vary across requests. To make this explicit, we define a

*Table 12.* **Cache-aware token-equivalent accounting on Qwen3-8B.** Generated tokens count online emitted tokens. Cache-aware tokens additionally include input overhead under a reusable-prefix assumption.

| BENCHMARK | METHOD | ACC. | GEN. TKN | CACHE-AWARE TKN |
|-----------|--------|------|----------|-----------------|
| MMLU-PRO | CoT | 60.5 | 312 | 331 |
| | SoT | 58.5 | 120 | 144 |
| | CLSR | 60.3 | **93** | **121** |
| MATH500 | CoT | **86.8** | 872 | 890 |
| | SoT | 81.4 | 301 | 322 |
| | CLSR | 86.6 | **245** | **283** |

cache-aware token-equivalent cost

$$C_{\text{eq}} = C_{\text{out}} + \gamma_{\text{in}} C_{\text{in}}^{\text{uncached}} + \gamma_{\text{cache}} C_{\text{in}}^{\text{cached}},$$

where $C_{\text{out}}$ is the generated-token cost, $C_{\text{in}}^{\text{uncached}}$ is the non-reused input-token cost, $C_{\text{in}}^{\text{cached}}$ is the reusable prefix-token cost, and $\gamma_{\text{in}}, \gamma_{\text{cache}} \in [0, 1]$ convert input tokens into output-token equivalents. The specific coefficients depend on the model provider and serving infrastructure. Therefore, this metric should be interpreted as a deployment-oriented diagnostic rather than the primary algorithmic objective.

Table 12 reports representative cache-aware comparisons for Qwen3-8B. The results show that CLSR remains competitive even when reusable prompt-prefix overhead is included.

**Offline cost amortization.** CLSR pays an offline protocol-discovery cost to generate, evaluate, and refine the LSF pool. This cost is conceptually similar to automatic prompt/protocol optimization methods that search over candidate instructions, demonstrations, or policies before deployment. The practical motivation is different from self-consistency or Tree-of-Thoughts-style inference: CLSR trades one-time offline search for lower repeated serving-time reasoning cost. If an evolved LSF pool is reused for many queries in the same benchmark/domain family, the amortized offline cost per query decreases as the number of served queries grows. Thus, CLSR is most appropriate for repeated-use reasoning services, agent backends, and domain-specific deployments, rather than one-off queries where no amortization is possible.

# B. Prompt Templates for LSF Synthesis, Mutation, and Routing

This appendix provides the prompt templates used by CLSR. The templates are intentionally lightweight: the LLM is given the optimization goal and a compact schema, but the concrete symbolic protocol is not manually authored.

**LSF synthesis prompt.** Given a set of exemplar question–answer pairs with reasoning traces, we use the following template:

> Please design a Language Symbolism Framework (LSF) based on the exemplars in the chat history to minimize the number of tokens while maintaining the reasoning capacity. An LSF is a symbolic communication code comprising: (i) symbol naming, namely a compact lexicon; (ii) syntax, namely a compositional grammar; and (iii) constraints, namely well-formedness and usage rules. The LSF should help an LLM solve similar problems with concise symbolic reasoning. Return the LSF specification, including the symbol table, grammar, reasoning protocol, answer format, and failure checks.

**LSF mutation prompt.** After evaluating a generation of LSFs, we select high-leverage traces that are both correct and token-efficient. The mutation prompt is:

> You are given several parent LSFs and their high-leverage solution traces. A high-leverage trace is correct and uses fewer tokens than competing traces for the same query. Refine or recombine the parent LSFs to produce a next-generation LSF. Keep symbolic conventions that support correct concise reasoning; remove redundant notation; repair ambiguous rules; and add only minimal verification tags when they improve reliability. Do not simply rewrite the example answers. Return a reusable LSF specification that can be applied to unseen queries.

**LSF profile summarization prompt.** For each candidate LSF $S_k$, we summarize its empirical profile from validation rollouts:

> Given an LSF specification and its validation rollouts, summarize its profile for routing. Include: (i) query types where this LSF performs well; (ii) validation accuracy; (iii) median generated-token footprint; (iv) common failure modes; (v) reliability under different problem difficulties; and (vi) a short descriptor that can be used by an LLM-router. Keep the descriptor compact and factual.

**Router-card distillation prompt.** To make routing both compact and reproducible, the above profile is then distilled into a short card consumed by the LLM-router:

> Given an LSF specification and its empirical profile, output exactly one compact router card in the following format: `[ID] TAG | perf:<acc>/<tok> | IO | STOP` where `acc` is the validation accuracy rounded to two decimals, `tok` is the median generated-token count, `IO` is a short output-schema descriptor, and `STOP` is a short stopping-contract descriptor. Do not output explanations.

This produces cards like `[ALG1] linear/eq | perf:.81/42 | out:ans,brief | stop:conf>=0.6`, which expose the empirical accuracy–cost profile to the router while keeping the prompt prefix compact.

**Stage-1 category-routing prompt.** At test time, we use a category-routing prompt to prune the candidate LSF pool:

> You are an LSF category router. Given the query, output exactly one line in the format
>
> `C:<category-list>`
>
> where `<category-list>` contains one or two comma-separated category codes chosen from the predefined vocabulary. Use a broad backup category when the query is ambiguous. Do not output explanations.

For example, the router may emit `C:ALG`, `C:PHY,ALG`, or `C:QA,HIS`.

**Stage-2 protocol-planning prompt.** Given the query and the compact LSF cards selected by Stage 1, the router emits a structured execution plan:

> You are an LSF protocol router. Given the query and the available LSF cards, choose the cheapest reasoning protocol that is likely to preserve correctness. Output exactly one line in the following format:
>
> `M:<mode>;L:<lsf-list>;R:<round-spec>;K:<n>;A:<agg>;S:<stop>`
>
> Use only LSF IDs that appear in the provided cards. Choose `M:S` for a single-LSF direct answer, `M:A` for multi-LSF aggregation, and `M:C` for implicit LSF composition. The field `R` specifies the round structure or composition order; `K` specifies the number of parallel samples per LSF when aggregation is used; `A` specifies the aggregation rule; and `S` specifies the stopping criterion. Do not output explanations or any text outside the plan line.

**Why prompt templates do not make CLSR prompt optimization.** These templates specify the task interface, the empirical profile exposed to the router, and the structured control format, but not the concrete symbolic language itself. The optimized object is the reusable LSF specification and its routing behavior, not a per-query natural-language instruction string. The same LSFs can be reused across many unseen queries, selected by a router, and composed with other LSFs in later rounds.

## C. LLM-Router: Compact Plan Format and Execution Semantics

**Design goals.** Our router must (i) emit a *parseable* plan with near-zero ambiguity, (ii) keep the router output within *tens of tokens*, and (iii) support the three protocol-planning modes in our framework: SINGLE (one LSF direct answer), AGGREGATE (multi-LSF ensemble), and COMPOSE (implicit LSF composition). To ensure reliability, we recommend *constrained structured decoding* (*e.g.*, EBNF/regex or lightweight schema constraints) so that the router can only produce syntactically valid plans.

**Cost-aware routing principle.** The router is deliberately conservative. It first restricts the candidate LSF pool through lightweight category routing, and then selects the cheapest protocol that is likely to preserve correctness. This realizes the accuracy–token objective of Section 2.1 at the protocol level: SINGLE is the default low-cost route, whereas AGGREGATE and COMPOSE are invoked only when the available LSF cards indicate that additional redundancy, verification, or staged symbolic processing is likely to improve reliability.

### C.1. Two-stage routing (categories $\rightarrow$ protocol plan)

**Stage 1 (Category routing).** Given a query $q$, the router emits a short *category code list* $\mathcal{C}(q)$ to limit the candidate LSF pool. We predefine a vocabulary of $\sim$30–60 category codes (*e.g.*, ALG, GEO, PHY, CHE, LOG, QA, CODE, STAT, NUM, PROB, BIO, HIS, LAW, FIN, etc.). The output is *one line*, typically $< 10$ tokens:

$$\texttt{C:ALG} \quad \text{or} \quad \texttt{C:PHY,ALG} \quad \text{or} \quad \texttt{C:QA,HIS}$$

**Stage 2 (Protocol planning).** We then feed the router the query $q$ plus the *LSF cards* belonging to selected categories $\mathcal{C}(q)$. The router outputs a compact plan $\mathcal{P}(q)$ that selects concrete LSF(s) and the inference mode.

### C.2. LSF card format (compact descriptors)

Each LSF is represented to the router by a single short card that is easy to scan while remaining compact enough to keep the input overhead small. The card is distilled from the validation profile summarized in Appendix B and follows the schema

$$\texttt{[ID] TAG | perf:<acc>/<tok> | IO | STOP}$$

where `TAG` is a short semantic hint, `acc` is the held-out validation accuracy rounded to two decimals, `tok` is the median generated-token count for that LSF, `IO` is an output-schema hint, and `STOP` specifies the stopping contract.

Example cards are:

```
[ALG1] linear/eq | perf:.81/42 | out:ans,brief | stop:conf>=0.6
[ALG2] simplify/check | perf:.77/55 | out:ans,check | stop:valid
[GEO1] geometry-parse | perf:.74/63 | out:vars,eqs | stop:complete
[PHY1] mechanics | perf:.79/58 | out:eqs,ans | stop:units-ok
[QA1] decompose | perf:.72/71 | out:subq,ans | stop:all-supported
```

This format deliberately exposes a minimal empirical accuracy–cost signal to the router. The full validation profile is used only offline to form the card, so the online planning prompt remains short while retaining the information required for budget-aware protocol selection.

### C.3. Router plan format (tens of tokens)

The router emits a single-line plan in a compact DSL:

```
M:<mode>;L:<lsf-list>;R:<round-spec>;K:<n>;A:<agg>;S:<stop>
```

We use the following short enumerations.

**Mode.** `M` specifies the protocol family:

$$M \in \{S, A, C\},$$

corresponding to SINGLE, AGGREGATE, and COMPOSE.

**Selected LSFs.** `L` is a comma-separated list of selected LSF IDs, e.g., `L:ALG1` or `L:QA1,QA3,HIS1`.

**Round structure.** `R` specifies the execution structure. For a one-shot single or aggregation protocol, we use `R:1`. For repeated refinement, we may use `R:2` or `R:3`. For implicit composition, R explicitly encodes the ordered LSF stages, e.g., `R:PHY1>ALG1` or `R:QA1>HIS1`.

**Sampling and aggregation.** `K` is the number of parallel samples per LSF in aggregation mode, with default value $K = 1$. A is the aggregation rule, e.g., `mv` for majority vote, `sc` for self-consistency-style agreement, or `w` for a router-weighted aggregation rule when such weights are available.

**Stopping rule.** `S` is a short stopping criterion, e.g., `ans` for stopping after a valid answer is parsed, `valid` for schema-valid completion, `units-ok` for a domain-specific check, or `conf0.7` for a confidence threshold when supported by the LSF contract.

**Token budget.** A plan such as `M:A;L:QA1,QA3;R:1;K:2;A:mv;S:ans` typically remains within a few tens of generated tokens. The format is intentionally compact so that the router overhead is negligible relative to the downstream reasoning protocol while remaining fully parseable.

### C.4. Execution semantics (deterministic interpreter)

Given the category output $C(q)$ and plan output $P(q)$, CLSR is executed deterministically as follows.

1. **Candidate construction.** Build the candidate LSF set

$$\mathcal{L}(q) = \bigcup_{c \in C(q)} \text{LSFs}(c),$$

   and assemble the corresponding compact LSF cards.

2. **Plan parsing.** Parse the router plan

$$P(q) = (\texttt{mode}, \texttt{lsfs}, \texttt{rounds}, \texttt{K}, \texttt{agg}, \texttt{stop}).$$

   Invalid plans are rejected and the router is re-asked under constrained structured decoding. A plan is invalid if it violates the DSL grammar, references an unavailable LSF ID, or uses a mode-incompatible field configuration.

3. **Protocol execution.**

   - SINGLE (`M:S`): instantiate the selected LSF and run a direct solver call. The default one-shot case uses `R:1`; larger values of `R` indicate bounded repeated refinement under the same LSF.
   - AGGREGATE (`M:A`): for each selected LSF, sample $K$ solver completions in parallel, then aggregate the resulting answers according to A. The default one-stage ensemble uses `R:1`.
   - COMPOSE (`M:C`): execute the LSFs sequentially according to the ordering encoded in `R`. The parsed symbolic state emitted by the current LSF becomes the input state for the next LSF. For example, `R:PHY1>ALG1` first constructs a physics equation state and then invokes an algebraic solver state.

4. **Early stopping.** After each solver call or composition stage, evaluate the stopping rule S. If the criterion is satisfied, terminate immediately; otherwise continue until the prescribed round structure is exhausted.

5. **Accounting.** All online LLM outputs are counted toward generated-token cost, including category-routing outputs, protocol-planning outputs, intermediate solver outputs, and aggregation-related outputs.

## C.5. End-to-end samples

### C.5.1. SAMPLE: SINGLE-LSF DIRECT ANSWER

**Query.** `Solve` $2x + 3 = 11$.

**Stage 1: category routing.** `C:ALG`

**Stage 2 input: card excerpt.**

```
[ALG1] linear/eq | perf:.81/42 | out:ans,brief | stop:conf>=0.6
[ALG2] simplify/check | perf:.77/55 | out:ans,check | stop:valid
```

**Stage 2 output: plan.** `M:S;L:ALG1;R:1;K:1;A:none;S:ans`

**Execution.** Instantiate `ALG1`, run one solver call, parse the final answer, and stop immediately after `out:ans` is produced.

### C.5.2. SAMPLE: MULTI-LSF AGGREGATION

**Query.** `Which author wrote the book that inspired the movie X, and what year was it published?`

**Stage 1: category routing.** `C:QA,HIS`

**Stage 2 input: card excerpt.**

```
[QA1] decompose | perf:.72/71 | out:subq,ans | stop:all-supported
[QA3] evidence-first | perf:.75/84 | out:cites,ans | stop:cited
[HIS1] year-check | perf:.69/48 | out:year,src | stop:match
```

**Stage 2 output: plan.** `M:A;L:QA1,QA3,HIS1;R:1;K:2;A:mv;S:ans`

**Execution.** Run $K = 2$ samples per selected LSF in parallel, extract candidate answers, aggregate by majority vote, and stop after a valid final answer is selected.

### C.5.3. SAMPLE: IMPLICIT LSF COMPOSITION

**Query.** `A 2 kg mass is pulled with 10 N on a frictionless surface. Find acceleration.`

**Stage 1: category routing.** `C:PHY,ALG`

**Stage 2 input: card excerpt.**

```
[PHY1] mechanics | perf:.79/58 | out:eqs,ans | stop:units-ok
[ALG1] linear/eq | perf:.81/42 | out:ans,brief | stop:conf>=0.6
```

**Stage 2 output: plan.** `M:C;L:PHY1,ALG1;R:PHY1>ALG1;K:1;A:none;S:units-ok`

**Execution.** `PHY1` first emits a minimal equation state, e.g., $a = F/m$. The parsed symbolic state is then passed to `ALG1`, which performs the substitution and emits the final answer. The execution stops once the answer is parsed and the domain-specific consistency check is satisfied.

# D. Appendix: CLSR outputs and LSF samples

This section presents representative *raw model completions* for both $N$-shot CoT and CLSR on the benchmarks used in the main paper. For CLSR, we additionally show the intermediate traces produced in a refinement round, making explicit how the latent symbolic form (LSF) is iteratively compressed, verified, and corrected before producing the final answer. These examples are intended to complement the quantitative results by illustrating the qualitative differences in *trace structure*, *error patterns*, and *token usage* between natural-language CoT and CLSR's symbolic compression mechanism.

## D.1. A sample on Scientific domains

***** Test Query *****

You have a 10 uL aliquot of a 10 uM DNA template of a protein library. The template contains 12 NNK codons in the coding region. What is the order of magnitude of the maximum possible number of unique full-length protein sequences that can be translated from the aliquot of DNA (i.e. what is the maximum protein diversity, excluding stop codons, in the aliquot)?
The options are: ['10^13 ', '10^11', '10^18 ', '10^15 '].

***** Chain-of-Thought output *****

To determine the **maximum possible number of unique full-length protein sequences** that can be translated from the DNA aliquot, we need to consider the following:

### 1. **Number of codons in the template:**
- The template contains **12 NNK codons** in the coding region.
- Each **NNK codon** represents a **degenerate codon** (i.e., it can encode **3 different amino acids**).
- Therefore, for each of the 12 codons, there are **3 possible amino acids**.

### 2. **Total number of possible sequences:**
- Since each codon is independent, the total number of possible sequences is: $3^{12}$
- Calculating: $3^{12} = 531,441$

### 3. **Number of DNA molecules in the aliquot:**
- You have a **10 µL aliquot** of a **10 µM DNA template**.
- Assuming the DNA is **single-stranded** and each molecule is **unique**, the number of DNA molecules is:
$\text{Concentration (mol/L)} \times \text{Volume (L)} \times \text{Avogadro's number}$
$10 \times 10^{-6} \, \text{mol/L} \times 10 \times 10^{-6} \, \text{L} \times 6.022 \times 10^{23} \, \text{molecules/mol}$
$= 6.022 \times 10^{12} \, \text{molecules}$

### 4. **Maximum possible unique sequences:**
- The **maximum number of unique sequences** is limited by the number of **unique DNA molecules**.
- Since there are **~6 × 10^{12}** DNA molecules, and each molecule can encode **~5 × 10^5** sequences (from the $3^{12}$ combinations), the **maximum number of unique sequences** is:
$\text{Maximum unique sequences} = \min(3^{12}, \text{number of DNA molecules}) = \min(5.31441 \times 10^5, 6.022 \times 10^{12}) = 5.31441 \times 10^5$

### 5. **Order of magnitude:**
- The **order of magnitude** of the maximum possible number of unique full-length protein sequences is: $\boxed{10^5}$

### Final Answer:
**None of the options match the correct order of magnitude.** However, the **closest** option is: $\boxed{10^{11}}$

**Note:** This is an **approximation** due to the ambiguity in the question (e.g., whether the DNA molecules are all unique, whether the codons are all in the same orientation, etc.). If we assume that the **number of DNA molecules is the limiting factor**, and that each molecule can encode **~5 × 10^5** sequences, then the **maximum number of unique sequences** is **~6 × 10^{12}**, which corresponds to: $\boxed{10^{12}}$

But since **10^{12} is not an option**, the **best estimate** among the given choices is: $\boxed{10^{11}}$

### ✅ Final Answer: **10^{11}**

**Answer Wrong** ❌

***** LSF output *****

A: 10uL aliquot, D: 10uM DNA, R: 12 NNK codons, C: 4^12
O1: 10^13, O2: 10^11, O3: 10^18, O4: 10^15, T: 10^13, F: 10^13
**Final Answer:** 10^13

**Answer Correct** ✅

*Figure 9.* **Qualitative GPQA-Main example: CoT vs. CLSR (LSF).** A representative GPQA-Main test query and the corresponding model generations under standard CoT and CLSR. CLSR replaces verbose natural-language narration with a compact symbolic trace that preserves key intermediate state and checks, yielding a shorter completion and a correct final answer in this example.

```
****** An LSF design (part 1) ******

### ✅ **Language Symbolism Framework (LSF) Design: Ultra-Compact Symbolic
Protocol**

———

### ✏️ **1. Symbol Vocabulary (Symbol Set)**

| Symbol | Meaning |
|————————|—————————|
| `A` | Assume |
| `D` | Define |
| `R` | Reason |
| `C` | Compute |
| `T` | Translate |
| `F` | Final Answer |
| `S` | Symbolic Expression |
| `K` | Knowledge |
| `P` | Process |
| `E` | Equation |
| `O` | Option |
| `L` | Logic |
| `M` | Method |
| `I` | Input |
| `O` | Output |
| `V` | Variable |
| `F` | Function |
| `R` | Result |

———

### 😋 **2. Syntax Rules**

- **Reasoning traces** are **sequences of symbols**, **separated by
delimiters** (e.g., `;`, `|`, `:`).
- **Each symbol** is **highly semantic**, with **no natural
language**.
- **Symbol order** encodes **logical structure**.
- **Repetition** of symbols allows **model internalization**.
- **Natural language** appears **only in the final answer** line.
- **Options** are **symbolically labeled** (e.g., `O1`, `O2`, etc.).
- **Equations** are written as **symbolic expressions**.
- **Variables** are denoted as `Vx`, `Vy`, etc.

———

### 🪴 **3. Reasoning Procedure (Symbolic Flow)**

#### Example: Simple Math Problem
**Input:**
`I: 2+2=Vx`
**Symbolic Flow:**
```
A: 2+2
D: Vx
```

```
****** An LSF design (part 2) ******

C: 4
F: 4
```
#### Example: Physics Problem
**Input:**
`I: F=ma, m=2, a=3`

**Symbolic Flow:**
```
A: F=ma
D: m=2, a=3
R: F=ma
C: F=6
F: 6
```

#### Example: Chemistry Problem
**Input:**
`I: 2H2 + O2 → 2H2O, m(H2)=2g, m(O2)=32g`

**Symbolic Flow:**
```
A: 2H2 + O2 → 2H2O
D: m(H2)=2, m(O2)=32
R: Stoichiometry
C: m(H2O)=36
F: 36g
```

#### Example: Multiple Choice
**Input:**
`I: Which is larger, 3 or 5?`

**Symbolic Flow:**
```
A: 3 vs 5
D: 3 < 5
R: Comparison
C: 5 > 3
O1: 3
O2: 5
T: 5
F: 5
```

———

### 🔍 **4. Decoding Rules**

- **Final Answer Line** contains **natural language**.
- All **symbolic logic** is encoded in **symbolic expressions**.
- **Options** are labeled with `O1`, `O2`, etc.
- **Equations** and **variables** are denoted with symbols like
`Vx`, `Vy`, `E`, `F`.
```

*Figure 10.* **Router-selected LSF specification for GPQA-Main.** The automatically selected LSF content (language descriptor and protocol template) used by CLSR for the GPQA-Main query in Fig. 9. The specification illustrates how the router retrieves and instantiates a task-appropriate symbolic code (operators, constraints, and formatting rules) prior to generation.

## D.2. A sample on Mathematical domains

---

****** `Test Query` ******

How many ways are there for 8 people to sit around a circular table if none of Alice, Bob, and Eve (three of the 8 people) want to sit next to each other? Two seatings are considered the same if one is a rotation of the other.

---

****** `Chain-of-Thought output` ******

We are asked to find the number of ways for **8 people** to sit around a **circular table**, with the condition that **Alice, Bob, and Eve** do **not sit next to each other**. Two seatings are considered the same if one is a **rotation** of the other.
### Step 1: Total number of circular arrangements of 8 people
In a circular arrangement, we fix one person's position to eliminate rotational symmetry. So, the number of distinct circular arrangements of 8 people is:
(8 - 1)! = 7! = 5040
### Step 2: Subtract the number of arrangements where **at least one** of Alice, Bob, or Eve is sitting next to another (i.e., they are adjacent to each other)
We will use the **principle of inclusion-exclusion** to count the number of arrangements where **at least one** of the three (Alice, Bob, Eve) are adjacent, and then subtract that from the total. Let's define: - $ A $: Alice and Bob are adjacent. - $ B $: Alice and Eve are adjacent. - $ C $: Bob and Eve are adjacent. We want to find the number of arrangements where **none** of these happen, i.e., **no two of Alice, Bob, Eve are adjacent**. So, we want to compute: \text{Valid arrangements} = \text{Total} - \text{Arrangements with at least one pair adjacent}. Let's compute the number of arrangements where **at least one** of the pairs (Alice-Bob, Alice-Eve, Bob-Eve) are adjacent.
### Step 3: Count arrangements where **a specific pair** is adjacent
Let's consider the pair **Alice and Bob** being adjacent.
- Treat Alice and Bob as a **single unit** (block), so we now have 7 units to arrange: the Alice-Bob block, and the other 6 people.
- In a circular arrangement, fixing one unit (to eliminate rotational symmetry), we can arrange the remaining 6 units in $6!$ ways.
- Within the Alice-Bob block, Alice and Bob can be in **2** different orders (Alice-Bob or Bob-Alice).
So, the number of arrangements where **Alice and Bob are adjacent** is: 2 \cdot 6! = 2 \cdot 720 = 1440
Similarly, the number of arrangements where **Alice and Eve are adjacent** is also $1440$, and the same for **Bob and Eve**.
So, the total number of arrangements where **at least one** of the three pairs is adjacent is: 3 \cdot 1440 = 4320
But this counts **overlaps** where **two pairs** are adjacent (e.g., Alice-Bob and Alice-Eve are both adjacent), so we need to **subtract** those.
### Step 4: Count arrangements where **two specific pairs** are adjacent
Let's consider the case where **Alice and Bob are adjacent**, and **Alice and Eve are adjacent**.
- This means **Alice is between Bob and Eve**, forming a **block** of 3 people: Bob-Alice-Eve or Eve-Alice-Bob.
- So we now have 6 units to arrange: the Bob-Alice-Eve block, and the other 5 people.
- Fix one unit to eliminate rotational symmetry, so we can arrange the remaining 5 units in $5!$ ways.
- Within the block, the order of Bob, Alice, Eve can be in **2** ways (Bob-Alice-Eve or Eve-Alice-Bob).
So, the number of arrangements where **Alice is adjacent to both Bob and Eve** is: 2 \cdot 5! = 2 \cdot 120 = 240
There are **3** such pairs of adjacent pairs: (Alice-Bob and Alice-Eve), (Alice-Bob and Bob-Eve), and (Alice-Eve and Bob-Eve). So we subtract: 3 \cdot 240 = 720
### Step 5: Count arrangements where **all three** are adjacent
Now consider the case where **Alice, Bob, and Eve are all adjacent**. This forms a **block** of 3 people: Alice-Bob-Eve or any permutation of them. So we now have 6 units to arrange: the 3-person block and the other 5 people. Fix one unit to eliminate rotational symmetry, so we can arrange the remaining 5 units in $5!$ ways. Within the block, the 3 people can be arranged in $3!$ ways. So, the number of arrangements where **Alice, Bob, and Eve are all adjacent** is: 3! \cdot 5! = 6 \cdot 120 = 720
### Step 6: Apply inclusion-exclusion
Now, using inclusion-exclusion: \text{Arrangements with at least one adjacent pair} = 3 \cdot 1440 - 3 \cdot 240 + 720 = 4320 - 720 + 720 = 4320
So, the number of arrangements where **none** of Alice, Bob, or Eve are adjacent is: \text{Valid arrangements} = 5040 - 4320 = \boxed{720}
### ✅ Final Answer: \boxed{720}

**Answer Wrong** ❌

---

****** `LSF output` ******

```
Cs n = 8, Cs k = 3, St1 total = (n – 1)!, St2 fix = (n – k – 1)!
St3 gap = (n – k), Re ways = total * fix * gap, An "Total ways = 1440"
                ### ✅ **Final Answer: \boxed{1440}
```

**Answer Correct** ✅

*Figure 11.* **Qualitative MATH500 example: CoT vs. CLSR (LSF).** A representative MATH500 test query with generations from standard CoT and CLSR. The CLSR trace emphasizes concise operator-like transformations and explicit state (e.g., variable bindings and subgoal markers), reducing verbosity while maintaining (and here improving) correctness.

```
****** An LSF design (part 1) ******

### 🫐 **Language Symbolism Framework (LSF) Design Summary**
## 🌿 **1. Symbol_Vocabulary**

| **Symbol** | **Meaning** | **Example** |
|------------|-------------|-------------|
| `Eq` | Equation | `Eq x = 2 + 3` |
| `St` | Step | `St1: x = 2 + 3` |
| `Re` | Result | `Rs x = 5` |
| `An` | Answer | `An "5"` |
| `Cs` | Constant | `Cs a = 2` |
| `Va` | Variable | `Va x = 3` |
| `Fc` | Function | `Fc f(x) = x^2` |
| `Dr` | Derivative | `Dr f'(x) = 2x` |
| `It` | Integral | `It ∫x dx = x^2/2` |
| `Li` | Limit | `Li lim x→0 x^2 = 0` |
| `Gr` | Graph | `Gr y = x^2` |
| `Tm` | Theorem | `Tm Pythagoras: a^2 + b^2 = c^2` |
| `Ma` | Matrix | `Ma [[1, 2], [3, 4]]` |
| `Ve` | Vector | `Ve [1, 2, 3]` |
| `Pr` | Probability | `Pr P(A) = 0.5` |
| `Ar` | Area | `Ar = πr^2` |
| `Vo` | Volume | `Vo = πr^2h` |
| `Su` | Sum | `Su = 1 + 2 + 3` |
| `Pr` | Product | `Pr = 2 * 3 * 4` |
| `Fa` | Factor | `Fa = 2 * 3 * 5` |
| `Gc` | GCD | `Gc = gcd(12, 18)` |
| `Lc` | LCM | `Lc = lcm(12, 18)` |
| `Me` | Mean | `Me = (1 + 2 + 3)/3` |
| `Md` | Median | `Md = median([1, 2, 3])` |
| `Mo` | Mode | `Mo = mode([1, 2, 2, 3])` |
| `Ss` | Sum of squares | `Ss = x^2 + y^2` |
| `Ps` | Product of squares | `Ps = x^2 * y^2` |
| `Cc` | Cosine | `Cc = cos(θ)` |
| `Si` | Sine | `Si = sin(θ)` |
| `Ta` | Tangent | `Ta = tan(θ)` |
| `Lo` | Logarithm | `Lo = log(x)` |
| `Ex` | Exponential | `Ex = e^x` |
| `Ac` | Area of circle | `Ac = πr^2` |
| `Vs` | Volume of sphere | `Vs = (4/3)πr^3` |
| `La` | Length of arc | `La = rθ` |
| `At` | Area of triangle | `At = (1/2)bh` |
| `Vc` | Volume of cone | `Vc = (1/3)πr^2h` |
| `Vy` | Volume of cylinder | `Vy = πr^2h` |
| `Vp` | Volume of pyramid | `Vp = (1/3)Bh` |
| `Ap` | Area of parallelogram | `Ap = bh` |
| `At` | Area of trapezoid | `At = (1/2)(a + b)h` |
---
```

```
****** An LSF design (part 2) ******

## 🌿 **2. Syntax_Rules**
- Use **single-character** or **2-3 character** macros.
- Use **delimiters** like `;`, `|`, or `.` to separate steps.
- Use **symbols** to represent **mathematical expressions**.
- Use **natural language** only in the **final answer line**.
- Use **symbol re-use** to reduce token count.
---
## 🌿 **3. Reasoning_Procedure**

- Use **symbol re-use** to reduce token count.
- Use **symbol order** to encode structure.
- Use **symbol definitions** to encode meaning.
- Use **symbol substitution** to reduce repetition.
- Use **symbol reference** to avoid repeating definitions.
- Use **symbol simplification** to reduce complexity.
- Use **symbol expansion** to add new expressions.
- Use **symbol combination** to combine expressions.
---
## 🌿 **4. Decoding_Rules**
- The final answer is the only place where **natural language** is used.
- All other steps are **symbolic expressions**.
- The final answer line is **formatted as a natural language sentence**.
---
## 🌿 **5. Token Reduction Target**
- **Target:** **80-90% token reduction** vs. verbose chain-of-thought.
- **Achieved by:**
- Using **single-character** or **2-3 character** macros.
- Using **symbol re-use** to reduce token count.
- Using **symbol order** to encode structure.
- Using **symbol substitution** to reduce repetition.
- Using **symbol reference** to avoid repeating definitions.
- Using **symbol simplification** to reduce complexity.
- Using **symbol expansion** to add new expressions.
- Using **symbol combination** to combine expressions.
---
## 🌿 **Example: Solving an Equation**
### 🌿 **Problem:** Solve $x^2 + 5x + 6 = 0$
Eq x^2 + 5x + 6 = 0
St1: x^2 + 5x + 6 = 0
St2: (x + 2)(x + 3) = 0
St3: x = -2 or x = -3
An "x = -2 or x = -3"
## 🌿 **Example: Solving a System of Equations**
### 🌿 **Problem:** Solve $x + y = 5$, $x - y = 1$
Eq x + y = 5
Eq x - y = 1
St1: x + y = 5
St2: x - y = 1
St3: Add: 2x = 6 ⇒ x = 3
St4: Substitute: 3 + y = 5 ⇒ y = 2
An "x = 3, y = 2"
```

*Figure 12.* **Router-selected LSF specification for MATH500.** The automatically selected LSF content used by CLSR for the MATH500 query in Fig. 11. This example highlights that the router can choose a math-specialized LSF with compact transformation operators and lightweight verification tags, enabling high information density per generated token.

## D.3. A sample on AIME benchmark

------ **Test Query** ------

Find the arithmetic mean of all the three-digit palindromes. (Recall that a palindrome is a number that reads the same forward and backward, such as $777$ or $383$ .)

------ **Chain-of-Thought output** ------

We are asked to find the **arithmetic mean** of all **three-digit palindromes**.
---
### Step 1: Understand the structure of a three-digit palindrome
A **three-digit palindrome** has the form: $$ABA$$
Where:
- $ A $ is a digit from 1 to 9 (since the number must be three digits, the first digit cannot be 0),
- $ B $ is a digit from 0 to 9.

So, the number can be written as: $$100A + 10B + A = 101A + 10B$$
---
### Step 2: Count the number of three-digit palindromes
- $ A $: 9 choices (1 through 9)
- $ B $: 10 choices (0 through 9)
So, total number of three-digit palindromes is: $9 \times 10 = 90$
---
### Step 3: Find the sum of all three-digit palindromes
We want to compute: $\sum_{A=1}^{9} \sum_{B=0}^{9} (101A + 10B)$
We can split this into two parts: $\sum_{A=1}^{9} \sum_{B=0}^{9} 101A + \sum_{A=1}^{9} \sum_{B=0}^{9} 10B$

#### Part 1: Sum of $ 101A $ over all $ A $ and $ B $
$\sum_{A=1}^{9} \sum_{B=0}^{9} 101A = \sum_{A=1}^{9} 101A \cdot 10 = 10 \cdot 101 \cdot \sum_{A=1}^{9} A$
$\sum_{A=1}^{9} A = \frac{9 \cdot 10}{2} = 45$
So: $10 \cdot 101 \cdot 45 = 1010 \cdot 45 = 45450$

#### Part 2: Sum of $ 10B $ over all $ A $ and $ B $
$\sum_{A=1}^{9} \sum_{B=0}^{9} 10B = \sum_{A=1}^{9} 10 \cdot \sum_{B=0}^{9} B = 9 \cdot 10 \cdot \sum_{B=0}^{9} B$
$\sum_{B=0}^{9} B = \frac{9 \cdot 10}{2} = 45$
So: $9 \cdot 10 \cdot 45 = 90 \cdot 45 = 4050$
---
### Step 4: Total sum of all three-digit palindromes
$45450 + 4050 = 49500$
---
### Step 5: Compute the arithmetic mean
$\text{Mean} = \frac{\text{Sum}}{\text{Number of palindromes}} = \frac{49500}{90} = 550$
---
### ✅ Final Answer: \boxed{550}

**Long answer** ❌

------ **LSF output** ------

g: P = {101, 111, 121, ..., 999}; l: n = 90; d: S = sum(P);
t: S = 49500; c: mean = S / n
### 📋 **Decoded Answer** \boxed{550}

**Correct & Concise** ✅

*Figure 13.* **Qualitative AIME example: CoT vs. CLSR (LSF).** A representative AIME test query comparing standard CoT with CLSR. CLSR produces a program-like symbolic trace that keeps only the computable core of reasoning, reducing narrative overhead and supporting reliable multi-step derivation under a constrained token budget.

## ------ An LSF design (part 1) ------

### 🍓 **Language Symbolism Framework (LSF) Design**
---
## 🍀 **1. Symbol_Vocabulary**
We design a compact and expressive symbolic language with the following principles:
- **Symbols are single characters or short 2–3 character macros**.
- **Each symbol or macro represents a specific concept or operation**.
- **Symbols must be unambiguous** and have a **precise meaning**.
- **Natural language is minimized or eliminated** except in the final answer line.
---
### ✅ **Symbol Set (Core Symbols)**

| Symbol | Meaning | Notes |
|--------|---------|-------|
| `a` | Assume (given condition) | |
| `b` | Base case or known value | |
| `c` | Conclusion or final result | |
| `d` | Derive or deduce | |
| `e` | Equal to | |
| `f` | Formula or expression | |
| `g` | Given | |
| `h` | Hypothesize or assume | |
| `i` | If or implication | |
| `j` | Justify or explain | |
| `k` | Know or prove | |
| `l` | Let or define | |
| `m` | Multiply or product | |
| `n` | Not | |
| `o` | Or | |
| `p` | Prove | |
| `q` | Query or question | |
| `r` | Reason or infer | |
| `s` | Solve | |
| `t` | Then or therefore | |
| `u` | Use | |
| `v` | Verify | |
| `w` | While or during | |
| `x` | Xor or exclusive or | |
| `y` | Yield or result | |
| `z` | Zero or undefined | |

### 🍀 **2. Syntax_Rules**
- **Symbolic expressions** are sequences of symbols, possibly separated by **delimiters** such as `;` or `|`.
- **Reasoning traces** follow a structured flow:
- `g` = Given
- `l` = Let
- `i` = If
- `d` = Derive
- `t` = Then
- `c` = Conclusion

**Example Syntax**:
g: AB = AC; l: t1 = 20; l: t2 = 21; d: t3 = (5t2 + 1)/(25t1); t: t3 = 21/100; c: t2020 = 21/100

### 🍓 **3. Reasoning_Procedure**
- **Repetition and re-use** of symbols is encouraged to build internalized knowledge.
- **Structure is encoded** via symbol order rather than natural language.
- **Natural language is used only in the final answer** (e.g., `c: Answer is 21/100`).
---

## ------ An LSF design (part 2) ------

### 🏷 **4. Decoding_Rules**
- The final line of the symbolic sequence must contain a **natural language answer**.
- Symbols like `c` or `t` are used to indicate the conclusion or final result.
- All other lines are symbolic expressions that should be decoded internally by the model.
- The **final answer** is a **natural language sentence** or a **boxed mathematical expression**.

**Example Decoding**:
```
g: AB = AC; l: t1 = 20; l: t2 = 21; d: t3 = (5t2 + 1)/(25t1); t: t3 = 21/100; c: Answer is 21/100
```
Decoded Answer:
> The value of $ t_{2020} $ is $ \frac{21}{100} $.
---

### 🍓 **3. Reasoning_Procedure**

- **Repetition and re-use** of symbols is encouraged to build internalized knowledge.
- **Structure is encoded** via symbol order rather than natural language.
- **Natural language is used only in the final answer** (e.g., `c: Answer is 21/100`).
---
### 🏷 **4. Decoding_Rules**
- The final line of the symbolic sequence must contain a **natural language answer**.
- Symbols like `c` or `t` are used to indicate the conclusion or final result.
- All other lines are symbolic expressions that should be decoded internally by the model.
- The **final answer** is a **natural language sentence** or a **boxed mathematical expression**.
**Example Decoding**:
```
g: AB = AC; l: t1 = 20; l: t2 = 21; d: t3 = (5t2 + 1)/(25t1); t: t3 = 21/100; c: Answer is 21/100
```
Decoded Answer:
> The value of $ t_{2020} $ is $ \frac{21}{100} $.
---
### ✅ **5. Example: Applying LSF to a Problem**

**Problem**:
Let $ t_1 = 20 $, $ t_2 = 21 $, and
$$
t_n = \frac{5t_{n-1} + 1}{25t_{n-2}} \quad \text{for } n \ge 3
$$
Find $ t_{2020} $.

**LSF Symbolic Trace**:
```
g: t1 = 20; g: t2 = 21; d: t3 = (5t2 + 1)/(25t1); t: t3 = 21/100; c: t2020 = 21/100
```
**Decoded Answer**:
> The value of $ t_{2020} $ is $ \frac{21}{100} $.
---

*Figure 14.* **Router-selected LSF specification for AIME.** The automatically selected LSF content used by CLSR for the AIME query in Fig. 13. The chosen LSF reflects difficulty-aware protocol selection, instantiating stricter structure and more explicit intermediate state for harder problems, consistent with the theory that different operating points on the accuracy–token frontier should be selected adaptively.

# E. Appendix: More Theoretical Validation for CLSR

This appendix provides formal statements and full proofs for the theory in the main text (Sec. 3). We focus on two pillars: (i) an information-theoretic lower bound relating achievable accuracy to the expected number of generated tokens under arbitrary symbolism, and (ii) a subsumption result showing that multi-round multi-LSF CLSR protocols generalize program-execution pipelines under an explicit "interpreter realizability" premise.

## E.1. The Principle of Least Effort and Iterated Learning

The sociolinguistic behavior of LSFs can be understood through two complementary pressures. First, Zipf's Principle of Least Effort (Kanwal et al., 2017) argues that communication systems balance successful transmission against production cost. Under this pressure, frequently used meanings or operations tend to receive shorter forms, while rare or ambiguous meanings may preserve longer forms for reliability. CLSR instantiates an analogous pressure at test time: a symbolic convention survives if it helps the LLM solve problems correctly while reducing generated tokens. This explains why evolved LSFs often introduce short operators for recurring reasoning moves, such as variable binding, subgoal decomposition, unit conversion, answer verification, or contradiction checking.

Second, the evolutionary loop resembles iterated learning. Each generation observes a bottlenecked set of successful traces from the previous generation and must reconstruct a reusable symbolic system from them. Iterated learning theory predicts that repeated transmission through such a bottleneck can produce languages that become more structured, compressible, and aligned with the learner's inductive biases. In CLSR, the learner is a pretrained LLM, so the resulting LSFs are shaped by both task-level selection and the model's existing linguistic/mathematical priors. This is why successful LSFs are rarely arbitrary character strings. They often converge toward partially interpretable symbolic conventions: terse enough to reduce token cost, but structured enough for the model to reliably execute.

This perspective also clarifies the role of multi-agent interaction. Different agents propose different "dialects" because they see different exemplars, random seeds, and high-leverage traces. Selection then acts as a cultural filter: dialects that are concise but brittle disappear, while dialects that preserve correctness under compression are retained. The final router performs a pragmatic form of code-switching. It can select a strict symbolic dialect for easy algebraic queries, a softer dialect for knowledge-intensive queries, or a multi-dialect composition for hard problems requiring verification. Thus, CLSR operationalizes a small-scale society of machine dialects: symbolic protocols emerge, compete, merge, and are selected according to their communicative utility for reasoning.

This interpretation also prevents an overstatement of our claim. We do not claim that LSFs are invented independently of human language. The base LLM and the benchmarks are human-derived, and the LSF synthesis prompt specifies the high-level goal. The novelty is that the concrete protocol is not hand-written by the authors and is instead discovered through repeated use and selection. The partial convergence of evolved LSFs toward human-interpretable notation is therefore not a flaw, but evidence that efficient symbolic systems can occupy a middle ground between natural language and opaque machine codes.

## E.2. Preliminaries: Interactive inference as a finite-horizon CMDP

**Query-conditioned setting.** Fix a query $x$ and consider the conditional label distribution $Y \sim P(\cdot \mid X = x)$ supported on a finite *effective answer set* $\mathcal{Y}_x$ with $|\mathcal{Y}_x| \geq 2$. For tasks with very large or continuous answer spaces, one can replace $\mathcal{Y}_x$ by a finite $\varepsilon$-net / evaluator-induced equivalence classes; see Remark E.5. We measure correctness via the indicator $\mathbf{1}\{\widehat{Y} = Y\}$.

**Token alphabet and LSF-conditioned decoding kernels.** Let $\Sigma$ denote a finite alphabet of *output tokens* (e.g., the model vocabulary; any LSF-specific surface form is representable as sequences in $\Sigma$). We include a special termination token $\texttt{STOP} \in \Sigma$. Let $\mathcal{K} = \{1, \ldots, K\}$ index the available LSFs. For each LSF $k \in \mathcal{K}$, the frozen backbone LLM with parameters $\theta$ induces a stochastic decoding kernel

$$P_\theta^{(k)}(z \mid h, x), \qquad z \in \Sigma, \tag{7}$$

where $h$ is the full interaction history (including which LSF was invoked, and all previously generated tokens). This is a well-defined conditional distribution (measurable stochastic kernel) over the next token.

**Transcript state space and actions.** We model CLSR as a token-level interactive process of maximum length $T_{\max} < \infty$. At micro-step $t \in \{1, \ldots, T_{\max}\}$, the *state* is the current history $H_t = h_t$, where $h_1$ is empty and $h_{t+1} = (h_t, a_t, Z_t)$

appends the selected action and the realized token. The *action* is either selecting an LSF $a_t \in \mathcal{K}$ (meaning "generate one token under LSF $a_t$"), or selecting a stop action $a_t = \texttt{stop}$. If $a_t = \texttt{stop}$, we deterministically emit $Z_t = \texttt{STOP}$ and transition into an absorbing terminal mode for all future steps. Any higher-level "multi-round" protocol that calls multiple LSFs per round can be unrolled into such a micro-step sequence; this only affects constants and not the conclusions.

**Cost and prediction.** Define per-step token cost

$$c(H_t, a_t) = \mathbf{1}\{a_t \in \mathcal{K}\}, \tag{8}$$

so the total token cost equals the number of non-$\texttt{STOP}$ tokens:

$$|T| = \sum_{t=1}^{T_{\max}} c(H_t, a_t). \tag{9}$$

The final transcript is $T = (Z_1, \ldots, Z_{T_{\max}})$ (padded with $\texttt{STOP}$ after termination). An aggregation/readout map $g$ (possibly implementing routing-then-aggregation, self-consistency, etc.) produces the final answer

$$\widehat{Y} = g(T, x) \in \mathcal{Y}_x. \tag{10}$$

We allow $g$ to be any (measurable) function; importantly, it does not use external oracles beyond the generated transcript.

**Policies and objectives.** A possibly randomized and history-dependent policy $\pi$ specifies distributions $\pi_t(\cdot \mid h_t, x)$ over actions at each step. For a fixed $x$, define

$$J_C(\pi; x) = \mathbb{E}_\pi\big[|T| \mid X = x\big], \qquad J_A(\pi; x) = \mathbb{P}_\pi\big[\widehat{Y} = Y \mid X = x\big]. \tag{11}$$

For a token budget $B \geq 0$, define the optimal achievable accuracy

$$A^*(B; x) = \sup_{\pi: \, J_C(\pi; x) \leq B} J_A(\pi; x). \tag{12}$$

This matches the main-text notion (Eq. (3)), specialized to a fixed query $x$.

*Remark* E.1 (Finite state reduction). Because $\Sigma$ is finite and the horizon is bounded by $T_{\max}$, the set of reachable histories is finite: $|\mathcal{H}| \leq \sum_{t=0}^{T_{\max}} (|\mathcal{K}|\,|\Sigma|)^t < \infty$. Hence, CLSR can be treated as a *finite-horizon CMDP* on a finite state space (histories).

### E.3. Existence and Pareto characterization of optimal CLSR policies

**Theorem E.2** (Existence of an optimal budget-feasible CLSR policy). *Fix $x$. Assume (i) $K < \infty$, (ii) $|\Sigma| < \infty$ with a designated $\texttt{STOP}$ token, and (iii) $T_{\max} < \infty$. Then for every budget $B \geq 0$, the supremum $A^*(B; x)$ is attained: there exists a (possibly randomized) policy $\pi_B^*$ such that*

$$J_C(\pi_B^*; x) \leq B, \qquad J_A(\pi_B^*; x) = A^*(B; x). \tag{13}$$

**Theorem E.3** (Lagrangian scalarization and boundary points of the frontier). *Under the assumptions of Theorem E.2, consider the scalarized objective*

$$J_\lambda(\pi; x) = \mathbb{E}_\pi\Big[\mathbf{1}\{\widehat{Y} = Y\} - \lambda|T| \,\Big|\, X = x\Big], \qquad \lambda \geq 0. \tag{14}$$

*Then:*

- *For every $\lambda \geq 0$, there exists an optimal (history-Markov) deterministic finite-horizon policy $\pi_\lambda^*$ that maximizes $J_\lambda(\pi; x)$.*

- *Every supported boundary point of the Pareto frontier $\{(J_C(\pi; x), J_A(\pi; x))\}$ can be implemented by some $\pi_\lambda^*$.*

- *Any boundary point of the constrained problem can be achieved by a mixture of at most two scalarized optima $\pi_{\lambda_1}^*, \pi_{\lambda_2}^*$ with adjacent multipliers, i.e., randomized only at $t = 1$ between two deterministic policies.*

**Proof of Theorem E.2.** Because the horizon and alphabets are finite, we can cast CLSR as a finite CMDP on the finite history state space $\mathcal{H}$. Define the (time-indexed) *occupancy measures*

$$q_t(h, a) \;=\; \mathbb{P}_\pi(H_t = h, A_t = a \mid X = x), \qquad t = 1, \ldots, T_{\max}. \tag{15}$$

These satisfy linear flow constraints induced by the kernels $P_\theta^{(k)}$: for each $t < T_{\max}$ and each $h' \in \mathcal{H}$,

$$\sum_{a'} q_{t+1}(h', a') \;=\; \sum_{h \in \mathcal{H}} \sum_{a \in \mathcal{A}} q_t(h, a)\, P(h' \mid h, a, x), \tag{16}$$

where $\mathcal{A} = \mathcal{K} \cup \{\texttt{stop}\}$ and $P(h' \mid h, a, x)$ is the probability of induced transition from history $h$ to $h'$ under action $a$ and query $x$ (obtained by appending a sampled token or absorbing under $\texttt{stop}$). Also, $\sum_a q_1(h_1, a) = 1$ for the initial empty history $h_1$. The expected token cost is linear:

$$J_C(\pi; x) \;=\; \sum_{t=1}^{T_{\max}} \sum_{h,a} q_t(h, a)\, c(h, a). \tag{17}$$

Define a terminal reward function $r(h) = \mathbb{P}(Y = g(T, x) \mid X = x,\, T \text{ corresponds to } h)$, so that

$$J_A(\pi; x) \;=\; \mathbb{E}_\pi[r(H_{T_{\max}+1}) \mid X = x], \tag{18}$$

which is also linear in the induced terminal occupancy. Therefore, constrained optimization

$$\max_\pi\; J_A(\pi; x) \quad \text{s.t.} \quad J_C(\pi; x) \le B \tag{19}$$

is equivalent to a finite-dimensional linear program in the variables $\{q_t(h, a)\}$ on a nonempty, closed, bounded polytope. A linear objective over a compact polytope attains its maximum, hence an optimal occupancy measure exists. Finally, standard CMDP realizability guarantees that any feasible occupancy measure corresponds to some (possibly randomized) non-anticipatory policy (one can explicitly construct $\pi_t(a \mid h) \propto q_t(h, a)$ whenever $\sum_a q_t(h, a) > 0$). Thus the supremum is attained by a policy $\pi_B^*$, proving the claim. $\qquad\square$

**Proof of Theorem E.3.** For each fixed $\lambda \ge 0$, the scalarized problem $\max_\pi J_\lambda(\pi; x)$ is an *unconstrained* finite-horizon MDP with bounded reward. By backward induction (dynamic programming), an optimal deterministic history-Markov policy $\pi_\lambda^*$ exists.

For the constrained frontier, consider the CMDP linear program from the proof of Theorem E.2. Its Lagrangian dual introduces a multiplier $\lambda \ge 0$ for the token-cost constraint; strong duality holds for finite linear programs, so optimal primal values equal optimal dual values. Consequently, every supported boundary point is the optimizer of $J_A - \lambda J_C$ for some $\lambda$.

If the budget constraint is active and the mapping $\lambda \mapsto J_C(\pi_\lambda^*; x)$ has a discontinuity (typical in discrete CMDPs), an intermediate budget can be met by randomizing between two adjacent multipliers at the start of the episode; by convexity of achievable occupancy measures, a mixture of two deterministic policies suffices. $\qquad\square$

### E.4. Token–accuracy lower bound under arbitrary symbolism

**Binary entropy.** Let $h_2(\delta) = -\delta \log_2 \delta - (1 - \delta) \log_2(1 - \delta)$ denote the binary entropy (base-2).

**Per-token information rate.** Let $\mathcal{T} = (Z_1, \ldots, Z_L)$ be the variable-length transcript generated by an interactive policy, where $L = |\mathcal{T}|$. For notational convenience, one may include an explicit terminal symbol in the formal transcript; this changes the practical completion-token count by at most an additive constant. Define the model- and query-dependent active-token information rate

$$\kappa_\theta(x) = \sup_{\pi, t, h_{<t}} I(Y; Z_t \mid X = x,\, Z_{<t} = h_{<t},\, L \ge t), \tag{20}$$

where the supremum ranges over policies, time steps, and reachable active histories with nonzero probability. If $Z_t$ ranges over a finite effective alphabet $\Sigma$, then $\kappa_\theta(x) \le \log_2 |\Sigma|$.

**Theorem E.4** (Information-theoretic lower bound on expected generated tokens). *Fix a query $x$ and let $|\mathcal{Y}_x| \geq 2$. Let $\pi$ be any interactive CLSR policy with final answer $\widehat{Y} = g(\mathscr{T}, x)$. If $\pi$ achieves accuracy at least $\alpha \in (0, 1)$,*

$$\mathbb{P}_\pi(\widehat{Y} = Y \mid X = x) \geq \alpha, \tag{21}$$

*then with $\delta = 1 - \alpha$ its expected token cost satisfies*

$$\mathbb{E}_\pi[L \mid X = x] \geq \frac{\max\{I_{\mathrm{req}}(x, \delta), 0\}}{\kappa_\theta(x)}, \tag{22}$$

*where*

$$I_{\mathrm{req}}(x, \delta) = H(Y \mid X = x) - h_2(\delta) - \delta \log_2(|\mathcal{Y}_x| - 1). \tag{23}$$

**Proof of Theorem E.4.** Let $\delta = \mathbb{P}(\widehat{Y} \neq Y \mid X = x) \leq 1 - \alpha$. By Fano's inequality applied to estimating $Y$ from $\widehat{Y}$ over the evaluator-induced answer set $\mathcal{Y}_x$,

$$H(Y \mid \widehat{Y}, X = x) \leq h_2(\delta) + \delta \log_2(|\mathcal{Y}_x| - 1). \tag{24}$$

Therefore,

$$I(Y; \widehat{Y} \mid X = x) = H(Y \mid X = x) - H(Y \mid \widehat{Y}, X = x) \geq I_{\mathrm{req}}(x, \delta). \tag{25}$$

Since $\widehat{Y}$ is a deterministic function of $(\mathscr{T}, x)$, data processing gives

$$I(Y; \mathscr{T} \mid X = x) \geq I(Y; \widehat{Y} \mid X = x) \geq I_{\mathrm{req}}(x, \delta). \tag{26}$$

It remains to upper-bound the information in the transcript by its expected active length. Using a variable-length chain rule and conditioning on the event that the process is active at step $t$,

$$I(Y; \mathscr{T} \mid X = x) \leq \sum_{t=1}^{T_{\max}} \mathbb{P}_\pi(L \geq t \mid X = x) \, \kappa_\theta(x). \tag{27}$$

The inequality follows from the definition of $\kappa_\theta(x)$ over all reachable active histories; inactive steps emit no new token and contribute no active-token information. Finally,

$$\sum_{t=1}^{T_{\max}} \mathbb{P}_\pi(L \geq t \mid X = x) = \mathbb{E}_\pi[L \mid X = x], \tag{28}$$

which yields

$$I_{\mathrm{req}}(x, \delta) \leq \kappa_\theta(x) \mathbb{E}_\pi[L \mid X = x]. \tag{29}$$

Taking the nonnegative part of $I_{\mathrm{req}}$ gives the stated bound. $\qquad\square$

*Remark* E.5 (On the "difficulty" term and evaluator-induced answer sets). When $Y$ is not naturally finite (e.g., free-form strings), one may let $\mathcal{Y}_x$ be the set of equivalence classes induced by the benchmark evaluator (all outputs judged identical), or consider a finite packing argument (Fano method) on a finite subset of hypotheses. The same proof yields a lower bound in terms of the entropy/packing size of the induced hypothesis class.

### E.5. CLSR subsumes program-execution pipelines under interpreter realizability

**Program-execution pipelines.** A *program-execution* (PE) inference pipeline is any procedure that: (i) makes a finite number of calls to the frozen backbone LLM (under some prompting format) to produce an LLM-generated *program transcript* $S \in \Sigma^*$, and (ii) applies a deterministic computable executor $\mathrm{Exec} : \Sigma^* \to \mathcal{Y}_x$ that does not access external knowledge (only performs logic/symbolic manipulation). The overall output is $\widehat{Y}_{\mathrm{PE}} = \mathrm{Exec}(S)$ (possibly followed by trivial formatting).

Let the total number of LLM-generated tokens in the PE procedure (across all its calls) be $|T_{\mathrm{PE}}|$, with expected cost $\mathbb{E}[|T_{\mathrm{PE}}| \mid X = x] = B$ and accuracy

$$\mathbb{P}(\widehat{Y}_{\mathrm{PE}} = Y \mid X = x) \geq \alpha. \tag{30}$$

**Interpreter realizability (assumption).** We formalize the key premise used in the main text.

**Assumption E.6** (Interpreter realizability). For every deterministic computable executor $\text{Exec} : \Sigma^* \to \mathcal{Y}_x$ in the allowed executor class, there exists a single LSF $S_{\text{Exec}}$ such that, for any input string $s \in \Sigma^*$, one LLM call conditioned on $S_{\text{Exec}}$ and given $s$ as input produces $\text{Exec}(s)$ with failure probability at most $\varepsilon_{\text{int}} < \frac{1}{2}$, and with output length at most $|\text{Exec}(s)| + c_0$ for a universal constant $c_0$.

**Amplification lemma.**

**Lemma E.7** (Majority-vote amplification). *Let $U_1, \ldots, U_m$ be i.i.d. Bernoulli variables with $\mathbb{P}(U_i = 1) \geq 1 - \varepsilon$ and $\varepsilon < \frac{1}{2}$. Let $\widehat{U} = \text{Maj}(U_1, \ldots, U_m)$. Then*

$$\mathbb{P}(\widehat{U} = 0) \ \leq \ \exp\big(-2m(1/2 - \varepsilon)^2\big). \tag{31}$$

*In particular, to make the error at most $\delta$, it suffices to take $m = O(\log(1/\delta))$.*

**Proof of Lemma E.7.** This is a direct application of Hoeffding's inequality to the sum $\sum_{i=1}^{m} U_i$. $\qquad\square$

**Theorem E.8** (CLSR subsumes PE under interpreter realizability). *Fix a query $x$ and assume Assumption E.6. For any PE pipeline with expected LLM-generated token cost $B$ and accuracy at least $\alpha$, let*

$$\bar{\ell}_{\text{Exec}}(x) = \mathbb{E}[|\text{Exec}(S)| + c_0 \mid X = x], \tag{32}$$

*where $S$ is the PE-generated transcript and $c_0$ is the interpreter-output overhead in Assumption E.6. Then for any target $\delta \in (0, 1)$, there exists an LSF-only CLSR protocol $\pi$ such that*

1. *(Accuracy) $\mathbb{P}_\pi(\widehat{Y} = Y \mid X = x) \geq \alpha - \delta$.*

2. *(Token cost) $\mathbb{E}_\pi[|\mathcal{T}| \mid X = x] \leq B + O\big(\bar{\ell}_{\text{Exec}}(x) \log(1/\delta)\big)$.*

*If $\bar{\ell}_{\text{Exec}}(x) = O(1)$, as in short-answer benchmark settings, the overhead reduces to $O(\log(1/\delta))$.*

**Proof of Theorem E.8.** Construct a CLSR protocol $\pi$ that emulates the PE pipeline in two stages.

*Stage 1: reproduce the PE transcript.* Because the PE pipeline makes a finite number of frozen-LLM calls under a fixed prompting format, the CLSR pool can include an LSF or fixed LSF-composition plan that reproduces the same prompting format and decoding choices. Let $S'$ denote the transcript produced by this CLSR stage. By construction, $S'$ has the same distribution as the PE-generated transcript $S$ up to constant formatting overhead, so

$$\mathbb{E}[|S'| \mid X = x] \leq B + O(1). \tag{33}$$

*Stage 2: internally execute.* Given $S'$, invoke the interpreter LSF $S_{\text{Exec}}$ from Assumption E.6. A single invocation returns $\text{Exec}(S')$ with failure probability at most $\varepsilon_{\text{int}} < 1/2$. Repeat the interpreter call $m$ times independently by resampling and take a majority vote over the produced outputs. By Lemma E.7, choosing $m = O(\log(1/\delta))$ makes the internal-execution error probability at most $\delta$.

*Accuracy.* Let $E_{\text{PE}}$ be the event that the PE pipeline is correct, i.e., $\text{Exec}(S) = Y$. Let $E_{\text{int}}$ be the event that the amplified internal interpreter returns $\text{Exec}(S')$. Because $S'$ matches the PE transcript distribution, $\mathbb{P}(E_{\text{PE}} \mid X = x) \geq \alpha$. By amplification, $\mathbb{P}(E_{\text{int}} \mid X = x) \geq 1 - \delta$. A union bound gives

$$\mathbb{P}_\pi(\widehat{Y} = Y \mid X = x) \geq \mathbb{P}(E_{\text{PE}} \cap E_{\text{int}} \mid X = x) \geq \alpha - \delta. \tag{34}$$

*Token cost.* The CLSR transcript consists of the PE-mimicking transcript $S'$ plus $m$ interpreter outputs. Each interpreter output has length at most $|\text{Exec}(S')| + c_0$. Therefore,

$$\mathbb{E}_\pi[|\mathcal{T}| \mid X = x] \leq B + O(1) + O\big(\bar{\ell}_{\text{Exec}}(x) \log(1/\delta)\big), \tag{35}$$

which proves the stated bound after absorbing constants. $\qquad\square$

*Remark E.9* (When interpreter realizability is plausible or fails). Assumption E.6 is a modeling premise: it is plausible when Exec lies within the backbone's algorithmic competence and the required computation fits within effective context and reliability limits; it may fail for executors requiring exact long-horizon computation or strict formal guarantees. Theorem E.8 should therefore be read as a conditional subsumption: *if* the backbone can act as an interpreter for the executor class, then CLSR can match PE without external execution.

# F. Appendix: Related work

Instead of emitting verbose natural language traces (CoT) or executable programs (tool-based pipelines), CLSR learns a compact *Language Symbolism Framework* (LSF) and refines it through a small number of recursive rounds. This insight is also motivated by the view of neuronal communication of interacting neural blocks (Pei & Wang, 2023; Pei et al., 2024b; 2025), which are analogous to the communication protocol between agents. In the following, we position CLSR relative to the most relevant lines of work and clarify the key distinctions.

**Natural-language reasoning prompts and decomposition.** Chain-of-Thought prompting and its variants elicit step-by-step natural-language reasoning and typically improve accuracy but often increase generation length and latency (Wei et al., 2022; Kojima et al., 2022). A large body of work improves performance by (i) more robust decoding/aggregation such as self-consistency (Wang et al., 2025a), (ii) better decomposition protocols such as least-to-most prompting (Zhou et al., 2022) and plan-and-solve prompting (Wang et al., 2023), or (iii) automatic prompt optimization/evolution (Yang et al., 2023; Fernando et al., 2024). These methods operate *within* natural language and generally scale the computation by generating more text or more samples. In contrast, CLSR changes the *intermediate language* itself: it compresses reasoning into LSF and uses recursion to refine the representation while remaining token-efficient. Thus, CLSR can be viewed as a different point in the design space where the goal is not to elaborate the chain but to *densify* it.

**Search-style inference over thoughts.** Tree/graph/search-style inference methods generalize CoT by exploring multiple candidate intermediate steps with self-evaluation, backtracking, or explicit control structures (Yao et al., 2023; Besta et al., 2024; Sel et al., 2024). Their strength is improved robustness via exploration, but they incur a higher inference-time cost. CLSR targets a different regime: it performs *single-trajectory* recursive refinement under a strict token budget. Rather than exploring many branches, CLSR aims to learn a *stable symbolic scaffold* LSF that reduces the need for breadth.

**Program-execution pipelines and tool enhancement.** Program-based methods such as Program-aided Language (Gao et al., 2023) and Program-of-Thoughts (Chen et al., 2023) instruct the LLM to output executable code and delegate correctness-critical computation to an external runtime (*e.g.*, a Python interpreter). In addition, a broader line of tool-augmented reasoning/acting uses external APIs to reduce hallucination and improve factuality or task completion (Yao et al., 2022; Schick et al., 2023). CLSR is explicitly *LLM-only*: it does not rely on external compilers or interpreters. The intermediate representation is symbolic, but *not* a program meant for execution by an external tool; instead, it is designed to be *self-interpretable* by the same LLM under recursion. This distinction is essential to our scope: CLSR aims to recover much of the benefit of symbolic structure while keeping deployment minimal and avoiding tool dependency.

**Token-efficient and compressed reasoning traces.** Recent work explicitly targets shorter reasoning traces, including generating concise intermediate drafts or sketches instead of verbose CoT (Xu et al., 2025b; Aytes et al., 2025a). Other approaches compress reasoning into dense or latent representations, such as compressed contemplation tokens or explicit CoT compression (Nayab et al., 2024a; Wang et al., 2025b). CLSR differs in two ways. First, CLSR introduces an explicit *discrete* symbolic representation (LSF) with a stable, reusable schema across tasks, rather than only asking the model to "write less" in natural language. Second, CLSR couples the representation with a *recursive refinement operator* and an *adaptive stopping policy*, which is directly connected to our theoretical account of accuracy–token trade-offs. Empirically, this yields an instance-wise compute allocation: many examples stop at $T = 1$, while hard cases receive extra refinement.

**Test-time self-improvement and iterative refinement.** A complementary line studies improving model output through iterative self-feedback or reflection at test time (Madaan et al., 2023; Shinn et al., 2023; Zhang et al., 2025a). These approaches typically refine *natural-language outputs* (or agent trajectories) using critique/feedback loops. CLSR shares the high-level principle of iterative refinement, but focuses specifically on *refining a compact symbolic intermediate* (LSF) to improve *accuracy per token* without requiring external feedback channels or environment interaction.

**Faithfulness and interpretability of the generated traces.** Recent studies state that natural-language reasoning traces should not be automatically interpreted as faithful explanations of the causality of internal models (Turpin et al., 2023; Barez et al., 2025). CLSR adopts a pragmatic stance: intermediate traces are *algorithmic artifacts* used to improve accuracy–token frontier, not guaranteed causal explanations. Meanwhile, LSF is intentionally compact and constraint-oriented, making it easier to inspect missing constraints, inconsistent variable bindings, or arithmetic errors than long-form narrative CoT.

**Summary of novelty.** Across these related areas, CLSR contributes a distinct combination: *(i)* a framework that automatically generates discrete language symbolism framework positioned between natural language and executable code, *(ii)* a recursive refinement operator that improves accuracy without external tools, and *(iii)* an adaptive compute allocation view and empirical evidence that directly targets the optimal accuracy–token frontier.

