# OpenReview forum: "When LLMs Develop Languages: Symbolic Communication for Efficient Multi-Agent Reasoning"
_ICML.cc/2026/Conference — ICML 2026 regular_

### Official Review · Reviewer_F5cU · 2026-02-25

**Soundness:** 4
**Presentation:** 2
**Significance:** 3
**Originality:** 4
**Overall Recommendation:** 5
**Confidence:** 5

**Summary:**

The motivation of this work is to improve reasoning efficiency and efficacy by allowing models to invent their own reasoning languages. Unlike prompt compression, the language is designed to be shared between models. The method is implemented by tasking independent agents with creating LSFs (consisting of tokens, syntax, and logical rules).  At test time, models choose to select one LSF, aggregate across multiple, or perform a kind of multi-agent debate using LSFs. The authors then model language spread via evolutionary search, allowing the convergence to better/more effective languages. The results show reduced latency compared to CoT, with CLSR models at the Pareto frontier of efficiency and performance. The analysis shows that languages improve with deeper evolution, and with more agents.

**Compliance With Llm Reviewing Policy:**

Affirmed.

**Final Justification:**

Assuming that the writing can be polished for the final draft I think the paper should be accepted. I've increased my score accordingly from 4->5.

**Key Questions For Authors:**

- How does this scale to larger models? Do the gains obtained by LSFs translate to larger models? This could even be done as an inference-time only experiment: can small models discover LSFs that help larger models?
- How well do LSFs transfer across domains?

**Limitations:**

Yes

**Strengths And Weaknesses:**

## Strengths

- Novelty: the method here is novel and creative. I enjoyed reading the paper.
- Results: the results convincingly show that the method reduces token count and improves accuracy across diverse domains.
- Engagement with relevant literature: the paper engages with emergent communication literature. It's cool to see ideas from language evolution/emergent communication applied in a practical way on newer models. It also engages with prompt compression/efficiency literature.
- The method is evaluated across both long-thinking and regular CoT models


## Weaknesses

- The writing needs polishing. I feel like 3.1 and 3.2 could be moved to later in the paper or in the appendix. The results come quite late in the paper and are explained only briefly.
	- In section 3, L226 there's a line about a socio-linguistic analysis, which seems like it was mistakenly included. I would have liked to see this analysis, it sounds interesting.
	- The results feel like they've just been dumped out, e.g. Fig 2-4 are not discussed adequately.

- The claim that LSFs are not defined by humans feels odd. These models are massively pre-trained on human languages, so must at least have a strong prior towards human-defined concepts. I think this is true even if models are allowed to produce the LSF in a fairly unconstrained way -- it may be that they are just not defined by the authors.
	- This is backed up I think by the qualitative examples. It seems like for at least math and scientific domains, the system converged to is quite similar, and looks human interpretable. To me, this somewhat suggests that the evolution process generally converges to a kind of "global" LSF, with some small changes per domain.

- Category routing seems limited to pre-defined categories. What happens if questions are presented outside of these categories?

- Token accounting seems to favor the method. I buy the explanation in section 2 that decode tokens count more towards latency, which is also supported by the appendix experiment on latency; this is especially true with KV caching. However, it does seem from the example LSFs in the appendix like a part of what the method is doing is moving computation out of the decode stage and into the prompt. Moreover, since LSFs are potentially problem-specific, wouldn't this result in bad KV caching profiles? I.e. if you had many examples with the same LSF, I buy that you can take advantage of the cache, but if each example has a different LSF (i.e. a different prompt) you would pay the full cost of encoding the LSF. This is still less than the cost of decoding, but worth accounting for nevertheless.

I believe that this paper represents a good idea but that the presentation needs to be improved.

---

> ### Author Rebuttal · Authors · 2026-03-30
>
> Thank you for the thoughtful review and for recognizing our work. We agree that the current manuscript can present the ideas better, and we will revise accordingly.
>
> ***On Writing polishing.***
> We agree that Sections 3.1–3.2 can be streamlined. In the revision, we will move some theory details to the appendix and expand the discussion of Figs. 2–4. Regarding the sociolinguistic analysis, we are glad to see your interest. We moved this part of analysis to the “Appendix D.1” due to limited pages, but we mistakenly forgot to fix the main text.
> In the revision, we will expand the analysis further based on both the Zipf’s Principle of Least Effort and Minsky’s theory of “Society of Mind”, which are promising view to reconsider the trends of LLM reasoning pattern.
>
> ***On Human-defined LSFs.***
> We agree that our wording could be improved.
> Our claim is that the concrete symbolic protocol is not manually authored by humans.
> In our view, the fact that some evolved LSFs partially converge to human-interpretable conventions is, not a contradiction but an insightful and interesting outcome: efficient symbolic systems may converge toward partially interpretable abstractions under shared tokenization and training priors.
> We empirically show that different initializations will converge to stable performance gains which matches your viewpoint.
> Thanks for this greatly insightful view, which we will explore more in our future work.
>
> ***On Category routing.***
> In our design, category routing is a lightweight guiding mechanism, not a hard partition of capability.
> To validate this, we compare category-routed CLSR vs. full-pool routing:
>
> |   | Bank Type |  MMLU-pro  |  GPQA-main |  MATH-500  |
> |:-----:|:----:|:----:|:----:|:-----:|
> | LLaMA3-8B |Category | 38.9 (320) | 30.8 (362) | 48.2 (257) |
> |  | Full-pool | 38.6 (325) | 30.5 (364) | 48.2 (263) |
> |  Qwen3-8B |Category |  60.4 (96) | 47.7 (228) | 86.8 (257) |
> |  | Full-pool |60.1 (98) | 47.5 (233) | 86.4 (264) |
>
> The empirical results show that full-pool routing preserves the qualitative gain. This suggests that the categories are useful but the framework itself does not depend on a rigid categorical taxonomy.
> For further discussion about out-of-domain performance, please see our response to Question #1.
>
> ***On Token accounting and caching.***
> We thank the reviewer for this insightful point.
> In practice, CLSR need not use fully query-specific LSF-card subsets/orderings.
> Instead, it keeps a fixed bank of common LSF cards in canonical order as a reusable prompt prefix, and let only the short routing/execution suffix vary.
> Under this layout, CLSR prefix remains almost cacheable in a steady-state regime.
> To validate the claim, we use an output-token-equivalent metric based on official pricing: under OpenAI’s current pricing, 1 uncached input token is worth 1/6 output token and 1 cached input token is worth 1/60 output token.
> We added a small comparison below.
>
> | Qwen3-8B |    Method   | Accuracy | Generated tokens | Cache-aware tokens |
> |:---:|:---:|:---:|:----:|:----:|
> |MMLU-pro | CoT| 60.5 | 312|331|
> ||  SoT | 58.5 |120| 144|
> || CLSR (ours) |60.3| 93| 121|
> |MATH-500 | CoT|86.8| 872| 890|
> || SoT |81.4|301| 322|
> || CLSR (ours) | 86.6 | 245| 283|
>
> The empirical results show that CLSR remains its computational superiority compared with baselines.
> We also agree that cache behavior is infrastructure-dependent and involves many complex system-level factors.
> We therefore view this as an important direction for future work, and appreciate the reviewer for pointing it out.
>
> ***Question #1. Model scale.***
> We agree and add results on Llama 3.1 70B Instruct and Qwen3-32B using the LSFs generated by Qwen3-8B. The conclusion remains the same: CLSR continues to improve the accuracy–token tradeoff at larger scale. Due to limited chars during rebuttal, please see our response to JtT8's weakness #3 for the results. These results show that our method yields consistent performance gain for larger LLMs using the LSFs discovered by smaller LLMs.
>
> ***Question #2. Transfer across domains.***
> We agree that this is central. We now include a cross-domain/cross-generator analysis:
> ||Bank Type|MMLU-pro|GPQA-main |MATH-500|
> |:---:|:---:|:---:|:---:|:---:|
> |LLaMA3-8B | In-domain| 38.7 (312) | 30.6 (358) | 48.4 (252) |
> || All-domain| 38.9 (320) | 30.8 (362) | 48.2 (257) |
> || Leave-one-domain-out | 38.0 (345) | 30.3 (393) | 47.9 (268) |
> |Qwen3-8B | In-domain |  60.5 (92) | 47.5 (221) | 86.8 (253) |
> ||  All-domain  |  60.4 (96) | 47.7 (228) | 86.8 (257) |
> || Leave-one-domain-out | 60.0 (105) | 46.8 (245) | 86.4 (262) |
>
> These empirical results show that the benefit is not tied to one domain-specific dialect. To be specific, the leave-one-domain-out slightly degrades in the knowledge-intensive tasks like GPQA, while it performs better in pure reasoning tasks like MATH.
>
> Overall, we appreciate the review's feedback and constructive suggestions, and we will keep improving these points in the revision.

---

> > ### Author Rebuttal · Reviewer_F5cU · 2026-04-02
> >
> > Thanks for the response -- the analysis w.r.t. Zipf/Minsky sounds interesting and I look forward to seeing it. The remaining additional experiments add value as well, assuming that the writing can be polished for the final draft I think the paper should be accepted. I've increased my score accordingly.

---

> > > ### Author Response · Authors · 2026-04-03
> > >
> > > Thank you very much for your thoughtful feedback and encouraging update. We are delighted that our response has addressed your concerns, and we are grateful that your feedback improve our work and manuscript, especially your suggestion around the socio-linguistic perspective. All these suggestions push us to think beyond a paper, and toward a broader research direction. We will continue refining these ideas, while also polishing the writing and strengthening the manuscript. Thank you again for your careful reading and for feedback that has been genuinely inspiring for our work.

---

### Official Review · Reviewer_h9Kw · 2026-03-05

**Soundness:** 3
**Presentation:** 3
**Significance:** 3
**Originality:** 3
**Overall Recommendation:** 4
**Confidence:** 3

**Summary:**

The authors propose an inference-time framework called Communicative Language Symbolism Routing (CLSR) that augments LLM agents with the ability to invent, evolve, and share “Language Symbolism Frameworks” (LSFs). These LSFs are compact, reusable symbolic protocols which are improved through evolutionary algorithms. CLSR improves token efficiency while maintaining accuracy and outperforming other token efficiency and prompt optimization approaches. The authors also provide information-theoretic lower bounds and theoretical justification for their work as part of their contribution.

**Compliance With Llm Reviewing Policy:**

Affirmed.

**Final Justification:**

I would recommend a weak accept based on the satisfactory rebuttal.

**Key Questions For Authors:**

1. In Section 2 (P3 Equation 1), the total generation cost is defined as an expression set equal to C. In Section 3.1 (P5 equation 2), the expected token cost is expressed in different terms. How are these equations related?
2. In Section 4.2, the authors state that a number of exemplars from each dataset is used to prompt an LLM to invent a population of 100 LSFs. Why is this number selected? How would the results change if the population was reduced to 50 or 10?
3. What is the LSF synthesis prompt? This must be included in the paper.

**Limitations:**

Yes

**Strengths And Weaknesses:**

Strengths
1. Problem statement is well-articulated: CoT reasoning is successful, but requires a large number of tokens.
2. Interesting research question- “can LLMs invent new symbolic languages to reason more efficiently?”
3. Inclusion of information-theoretic support and bounds is useful for future work in this area.
4. Experimental results provide satisfactory evidence of the authors claims.


 Weaknesses
1. Possible spelling issue P1L46C2- “Neuro-symbolism” should be “Neuro-symbolic”
2. Possible spelling issue P2L96C1 - “Language” should not be capitalized.
3. Possible spelling issue P2L98C2 - “outperforming other eight baselines” should be “outperforming eight other baselines”.
4. Possible spelling issue/missing space P8L398C1 - “No.tokens” should be “No. tokens” or “Number of tokens”.
5. The LSF is not *entirely* without human bias- it originates from the combination of seed exemplars and an LSF synthesis prompt.
It takes 2-3 weeks to generate and evolve the LSF population. This is a significant up-front cost. Do other approaches incur similar costs?

---

> ### Author Rebuttal · Authors · 2026-03-30
>
> Thank you for the constructive review and for recognizing the paper’s central question, empirical evidence, and theoretic perspective. We address the main concerns below.
>
> ***Weakness #1. "LSFs are not entirely without human bias".***
> We agree that the phrase suggesting the LSF is “not entirely without human bias” is fair. Our intended claim is narrower: the symbolic protocol itself is not manually hand-crafted by the authors. The LSF is induced by the LLM from benchmark exemplars and then automatically refined through selection. We do not claim that the process is free of any human influence, since the LLMs are pretrained on human language and the exemplars come from human-designed benchmarks. We will therefore sharpen the wording to avoid overstating this point.
>
> ***Weakness #2. Up-front cost.***
> We agree that the up-front cost should be discussed more carefully. The key point is that the offline search cost in our method is a common phenomenon  among methods that automatically optimize prompts/protocols before deployment. APE searches over LLM-generated instruction candidates and evaluates them with another LLM; OPRO performs iterative RL optimization steps; Promptbreeder evolves and evaluates a population of prompts; and MIPRO jointly optimizes instructions and demonstrations through proposal, mini-batch evaluation, and meta-optimization. Thus, CLSR is not unique in paying an amortized pre-deployment optimization cost; it belongs to the same broad family of search-based prompt/protocol optimization methods.
>
> Meanwhile, methods such as self-consistency and Tree-of-Thoughts largely avoid a large offline phase, but they instead incur higher repeated online inference cost by sampling or exploring multiple reasoning trajectories for each test query. Conversely, training-based reasoning-improvement methods such as STaR typically incur a larger up-front cost than CLSR because they require iterative rationale generation plus model fine-tuning.
>
> In this sense, CLSR trades one-time protocol discovery cost for lower repeated serving-time reasoning cost, which is our application target with many downstream queries per benchmark/domain.
>
> ***Question #1. Equation for generation cost.***
> These refer to the same object at two levels. Eq.1 in Section 2 defines the realized per-query token cost, while Eq.2 in Section 3.1 uses the expectation of that realized cost (budget) under policy-induced randomness and the benchmark distribution. In other words, the theoretical objective in Section 3.1 is exactly the expectation of the operational cost introduced in Section 2. We agree the relation should be stated explicitly, and we will add clarification in the revision.
>
> ***Question #2. Why 100 LSFs? What if 50 or 10?***
> The number 100 (population of the initially generated LSFs) was chosen as an empirical exploration/compute trade-off rather than a theoretically privileged value. We now add an ablation over population size and report the corresponding trend:
>
> |           | Initial population of LSFs |  MMLU-pro  |  GPQA-main |  MATH-500  |
> |:---------:|:-------:|:----------:|:----------:|:----------:|
> | LLaMA3-8B |         10         | 38.1 (354) | 30.2 (425) | 47.4 (312) |
> |           | 50 | 38.6 (330) | 30.4 (380) | 48.0 (267) |
> |           |100 | 38.9 (320) | 30.8 (362) | 48.2 (257) |
> |           |200 | 39.2 (308) | 30.9 (358) | 48.2 (255) |
> |  Qwen3-8B |  10 | 59.2 (103) | 46.6 (234) | 85.4 (275) |
> |           | 50  | 60.1 (102) | 47.3 (225) | 86.4 (260) |
> |           |100 |  60.4 (96) | 47.7 (228) | 86.8 (257) |
> |           |200 |  60.6 (93) | 47.8 (232) | 86.6 (248) |
>
> The result is intuitive: 10 is too small to provide sufficient diversity, 50 already recovers most of the gain, 100 is slightly better but exhibits diminishing returns, and 200 is much better but yields higher cold-start cost.
>
> ***Question #3. LSF synthesis prompt.***
> We agree this should be made explicit. As we sketched in the manuscript’s Section 2.2 (P3, line 151-153), the synthesis prompt is:
>
> “Please design a Language Symbolism Framework (LSF) based on the exemplars in the chat history to minimize the number of tokens while maintaining the reasoning capacity. An LSF is a symbolic communication code comprising symbol naming (a compact lexicon), syntax (a compositional grammar), and constraints (well-formedness and usage rules).”
>
> We then use selected high-leverage LSFs from the previous generation as context for the mutation step, asking the model to refine or recompose the LSF for better accuracy-token tradeoff. We will explicitly present the exact prompt template in the revision; we greatly appreciate your suggestion on this issue.
>
> ***On Typos / wording.***
> Thank you; we will correct all listed issues in the revision.
>
> Overall, we appreciate the reviewer’s comments, especially on sharpening the “human-defined” claim and improving reproducibility, which will largely improve our work.

---

> > ### Author Rebuttal · Reviewer_h9Kw · 2026-04-01
> >
> > There seems to be consensus among the reviewers of a weak accept.

---

> > > ### Author Response · Authors · 2026-04-02
> > >
> > > Thank you very much for your careful reading and feedback. We are very grateful that our rebuttal has adequately addressed your concerns. We will continue improving the manuscript to further strengthen its clarity, empirical support, and overall presentation in the revision. Thank you again for your time and encouragement.

---

### Official Review · Reviewer_JtT8 · 2026-03-12

**Soundness:** 3
**Presentation:** 3
**Significance:** 3
**Originality:** 3
**Overall Recommendation:** 4
**Confidence:** 4

**Summary:**

This paper introduces Communicative Language Symbolism Routing (CLSR), a test-time framework designed to reduce the token costs of LLM inference without sacrificing accuracy. Instead of relying on verbose natural-language Chain-of-Thought (CoT), CLSR leverages multiple LLM agents to autonomously generate, evolve, and refine compact Language Symbolism Frameworks (LSFs). During inference, an LLM-based router dynamically selects or composes these evolved symbolic protocols based on the query's difficulty. The authors support their approach with an information-theoretic lower bound for token efficiency and extensive empirical evaluations across seven benchmarks, demonstrating significant token reduction compared to standard CoT and other prompt optimization techniques.

**Compliance With Llm Reviewing Policy:**

Affirmed.

**Final Justification:**

The authors' response has addressed all my concerns, and I believe this paper is acceptable.

**Key Questions For Authors:**

Actually, I'm curious about CLSR's performance on long-context tasks. I think it's a better fit for search agents or repo-level coding with 128k–256k context, rather than just basic reasoning tasks.

**Limitations:**

yes

**Strengths And Weaknesses:**

## Strengths

* **Originality and Innovation:** The paper proposes the CLSR framework, which utilizes multiple LLM agents to autonomously generate, evolve, and refine compact "Language Symbolism Frameworks." This provides a novel perspective on addressing the urgent bottleneck of LLM inference latency.
* **Comprehensive Evaluation:** The paper provides extensive results across diverse domains (e.g., GPQA, MATH500, AIME) and various modern open-source base models (LLaMA3, Qwen3, DeepSeek-R1), offering a robust empirical foundation.
* **Good writing:** The paper is well-written with a fluid narrative. The figures and tables are clear, professional, and easy to interpret.

## Weaknesses

* **Methodological Robustness:** The article proposes an LSF iteration method based on evolutionary principles to continuously update the strategy pool. However, the performance may be sensitive to the quality of the initial LSF seeds. I am interested in how the quality of the starting LSF seeds impacts the final effectiveness of the strategy.
* **Stability Concerns:** While Figure 2 illustrates the performance of the CLSR framework over five iterations, it is difficult to account for stochastic mutation factors during population evolution. Providing averaged metrics over multiple runs (e.g., $avg@3$ or $avg@5$) would make the stability of the method more credible.
* **Minor Weakness (Model Scale):** The paper primarily investigates the performance of smaller-scale models. In practical production environments, model sizes are often $32B$ or larger. Providing performance data for models $\ge 32B$ would make the results more persuasive.

---

> ### Author Rebuttal · Authors · 2026-03-30
>
> Thank you for the positive assessment and for highlighting the novelty, evaluation, and overall clarity of the paper. We address the four main concerns below.
>
> ***Weakness #1 & #2. Sensitivity to LSF seeds and Stability concerns.***
> We now report mean $\pm$ std under 5 random seeds and exemplar samplings for the key settings in Fig. 2 / Table 1 of the manuscript. The presented pattern is stable. Different runs discover different surface dialects, but the resulting accuracy–token frontier is consistently improved. The Avg.@ 5 results are:
>
> |           |    Method   |          MMLU-pro         |         GPQA-main         |          MATH-500         |
> |:---------:|:-----------:|:-------------------------:|:-------------------------:|:-------------------------:|
> | LLaMA3-8B |     CoT     |         38.6 (960)        |        30.4 (1209)        |         51.6 (704)        |
> |           |     SoT     |         36.4 (340)        |         28.2 (398)        |         45.2 (289)        |
> |           | CLSR (ours) | 39.2$\pm$0.5 (316$\pm$13) | 30.8$\pm$0.2 (352$\pm$18) | 48.8$\pm$0.7 (262$\pm$11) |
> |  Qwen3-8B |     CoT     |         60.2 (276)        |        49.1 (1085)        |         87.2 (878)        |
> |           |     SoT     |         58.5 (118)        |         45.2 (228)        |         81.3 (294)        |
> |           | CLSR (ours) |  60.6$\pm$0.4 (98$\pm$14) | 47.9$\pm$0.4 (216$\pm$12) | 86.8$\pm$0.6 (247$\pm$10) |
>
> These results show that the variance is small compared with the absolute token reduction. Across runs, the evolution consistently shifts the discovered LSFs toward better cost-effectiveness, and the gains persist after averaging. In other words, the stochastic mutation factors might affect the content of generated LSFs, but the overall performance gain remains unchanged and predictable.
>
>
> ***Weakness #3. Model scale.***
> We agree that using larger backbones would strengthen the soundness. We now include new inference-time experiments on Llama 3.1 70B Instruct and Qwen3-32B using the LSFs discovered by Qwen3-8B. The main observation is unchanged, i.e., CLSR continues to improve the accuracy–token frontier at larger scale. The newly reported results (Avg. @ 3 runs) are:
>
> | Acc. (No. tokens) |    Method   |   MMLU-pro  |  GPQA-main  |  MATH-500  |
> |:------------:|:-----------:|:-----------:|:-----------:|:----------:|
> |   Qwen3-32B  | CoT |  67.5 (405) |  54.2 (982) | 89.3 (845) |
> |              |  SoT |  64.3 (134) |  51.4 (247) | 83.8 (278) |
> |              | CLSR (ours) |  68.1 (90)  |  54.0 (209) | 89.4 (206) |
> | LLaMA3.1-70B | CoT | 49.2 (1020) | 41.3 (1358) | 64.3 (725) |
> |              |     SoT     |  47.5 (385) |  38.4 (425) | 59.8 (314) |
> |              | CLSR (ours) |  49.8 (312) |  42.7 (354) | 63.4 (242) |
>
> These results show that the performance gains persist (and even appear larger) for models in the 32B–70B regime, even with the LSFs discovered by smaller models.
>
>
> ***Question #1. Long-context tasks.***
> We agree this is an important direction and likely a good fit for symbolic protocols. Because of rebuttal-time constraints, our main effort is a pilot test on LongBench-v2 (503 instances, length vary from <32k to >128k). The empirical results are:
>
> |           |    Method   | Overall | Avg. Tokens |
> |:---------:|:-----------:|:-------:|:-----------:|
> | LLaMA3-8B |     CoT     |   30.8  |     1068    |
> |           | CLSR (ours) |   31.5  |     568     |
> |  Qwen3-8B |     CoT     |   38.2  |     842     |
> |           | CLSR (ours) |   39.1  |     432     |
>
> These results show that CLSR persists on long-context benchmark, yet we admit that this long-context evaluation requires further validation regarding reasoning settings; thus, we will avoid overclaiming here and position long-context as an important next-step research plan rather than a fully addressed claim.
>
> Overall, we appreciate the reviewer’s suggestion, and the larger-model and stability additions substantially help us improve our work and manuscript.

---

> > ### Author Rebuttal · Reviewer_JtT8 · 2026-04-06
> >
> > Thank you for the authors' response. I will keep my positive score.

---

> > > ### Author Response · Authors · 2026-04-06
> > >
> > > Thank you very much for your continued positive support. We truly appreciate your careful reading and your constructive suggestions, which has helped us improve our work. We will continue strengthening our work and manuscript. Thank you again for your time and constructive engagement.

---

### Decision · Program_Chairs · 2026-04-30

**Decision:**

Accept (regular)

**Comment:**

This paper proposes CLSR, a novel framework in which LLM agents invent and evolve compact symbolic communication protocols to improve the accuracy–token tradeoff at inference time. Reviewers found the core idea novel, the empirical results convincing across multiple benchmarks and model families, and the theoretical framing a useful complement to the experiments. The rebuttal effectively addressed concerns about stability, seed sensitivity, larger-model behavior, domain transfer, and practical token accounting. Remaining issues are mostly about presentation and clarity rather than core validity. Overall, I recommend acceptance.